# A protein-specific priority code in presequences determines the efficiency of mitochondrial protein import

Saskia Rödl[1], Yasmin Hoffman[1], Felix Jung[2], Annika Egeler[1], Annika Nutz[1], Oliver Šimončík[3], Martin Jung[4], Markus Räschle[5], Petr Muller[3], Zuzana Storchová[5], Timo Mühlhaus[2], Johannes M. Herrmann[1]*

1 Cell Biology, University of Kaiserslautern, RPTU, Kaiserslautern, Germany, 2 Computational Systems Biology, University of Kaiserslautern, RPTU, Kaiserslautern, Germany, 3 Masaryk Memorial Cancer Institute, RECAMO Research Centre for Applied Molecular Oncology, Brno, Czech Republic, 4 Medical Biochemistry and Molecular Biology, Saarland University, Homburg, Germany, 5 Molecular Genetics, University of Kaiserslautern, RPTU, Kaiserslautern, Germany

* Hannes.herrmann@biologie.uni-kl.de

## Abstract

The biogenesis of mitochondria relies on the import of hundreds of different precursor proteins from the cytosol. Most of these proteins are synthesized with N-terminal presequences which serve as mitochondrial targeting signals. Presequences consistently form amphipathic helices, but they considerably differ with respect to their primary structure and length. Here we show that presequences can be classified into seven different groups based on their specific features. Using a test set of different presequences, we observed that group A presequences endow precursor proteins with improved in vitro import characteristics. We developed IQ-Compete (for Import and de-Quenching Competition assay), a novel assay based on fluorescence de-quenching, to monitor the import efficiencies of mitochondrial precursors in vivo. With this assay, we confirmed the increased import competence of group A presequences. Using mass spectrometry, we found that the presequence of the group A protein Oxa1 specifically recruits the tetratricopeptide repeat (TPR)-containing protein TOMM34 to the cytosolic precursor protein. TOMM34, and the structurally related yeast co-chaperone Cns1, apparently serve as presequence-specific targeting factors which increases the import efficiency of a specific subset of mitochondrial precursor proteins. Our results suggest that presequences contain a protein-specific priority code that encrypts the targeting mechanism of individual mitochondrial precursor proteins.

## Introduction

Mitochondria are essential organelles of eukaryotic cells that play a crucial role in metabolism and ATP synthesis. They consist of 900 (yeast) to 1,500 (animals) proteins which, with few exceptions, are synthesized in the cytosol [1,2]. Most proteins of

**Data availability statement:** All mass spectrometry proteomics data are available from the ProteomeXchange Consortium via the PRIDE partner repository with the dataset identifiers PXD053210 and PXD059975. The flow cytometry data is deposited in Zenodo: https://doi.org/10.5281/zenodo.15740860

**Funding:** This work was supported by the Deutsche Forschungsgemeinschaft (HE2803/11-1 to JMH and STRESSistance to JMH, TM and ZS) (https://www.dfg.de/de), the European Research Council (MitoCyto 101052639 to JMH) (https://erc.europa.eu/homepage), the Research Initiative of Rheinland-Pfalz (BioComp to ZS, TM and JMH) (https://mwg.rlp.de/themen/wissenschaft/forschung/forschung-an-hochschulen/forschungsinitiative), the European Union and the State Budget of the Czech Republic [project SALVAGE, P JAC; CZ.02.01.01/00/22_008/0004644 to PM], the Czech Science Foundation (22-17102S to PM) and the Ministry of Health Development of Research Organisation, MH CZ - DRO (MMCI, 00209805 to PM). The sponsors or funders didn't play a role in the study design, data collection and analysis, decision to publish, or preparation of the manuscript.

**Competing interests:** The authors have declared that no competing interests exist.

**Abbreviations:** BSA, bovine serum albumin; DHFR, dihydrofolate reductase; GFP-Q-D, GFP-quencher-degron; GO, gene ontology; LFQ, label-free quantification; MRPs, mitoribosomal proteins; MTS, mitochondrial targeting signals; PK, proteinase K; TPR, tetratricopeptide repeat; UMAP, Uniform Manifold Approximation and Projection; uTEV, Tobacco Etch virus protease; YEP, yeast extract peptone.

the mitochondrial matrix and the inner membrane are made with N-terminal presequences which serve as mitochondrial targeting signals (MTS) directing the precursor proteins through the translocases of the mitochondrial import machinery (for overview see [3–5]).

These presequences form amphipathic helices with one hydrophobic and one positively charged surface, are rich in hydroxylated but devoid of negatively charged residues, and are removed in the mitochondrial matrix by proteolytic cleavage by the matrix processing peptidase MPP [6,7]. Their common structural features allow robust detection by rather simple prediction programs, such as TargetP [8], MitoFates [9], TPpred3 [10], and DeepMito [11]. These algorithms are based on machine-learning approaches that were trained with proteins of known location and can reliably distinguish mitochondrial from non-mitochondrial proteins of yeast and animal cells. However, it should be noted that these programs rely on the presence of N-terminal MTS, and thus do not recognize targeting signals in proteins of the mitochondrial outer membrane and in many inner membrane and intermembrane space proteins which typically lack presequences.

Despite their structural similarity, presequences strongly differ in length, ranging from less than 10 to more than 100 residues [7,12]. Presequences are both necessary and sufficient for mitochondrial targeting; when fused to the N-terminus of any given protein sequence they reliably target it to the matrix of mitochondria unless tightly folded domains prevent membrane translocation [13,14]. Mechanistically, presequences act at different levels; their primary function is to mediate the binding of precursors to the mitochondrial surface. Tom20, potentially in conjunction with Tom22, acts as a cytosol-exposed presequence receptor of the translocase of the outer membrane, the TOM complex [5,15–18]. Tom70, that is less tightly associated with the TOM complex, provides a secondary binding site which can take over if Tom20 is absent, but also might gather precursors on the mitochondrial surface and usher them to Tom20. Moreover, Tom70 binds to internal presequence-like segments in the mature part of mitochondrial precursors, so-called iMTS sequences, and interacts with cytosolic chaperones of the Hsp70 and Hsp90 classes as well as with J proteins [19–24]. Therefore, Tom70 serves as an interface between the cytosolic proteostasis network and the mitochondrial import machinery.

However, presequences exhibit further functional roles, including the binding of cytosolic chaperones that maintain precursors in an import-competent state [25–27], serving as degradation signals in case of import failure [28–30], promoting precursor insertion into the protein-conducting channel of the translocase of the TOM complex and the presequence translocase of the inner mitochondrial membrane, the TIM23 complex, and serving as an accession sequence for the import motor and matrix Hsp70 chaperones which ratchet precursors into the organelle [31–33]. It remains unknown whether each of these functions are conjointly carried out by the structural features of presequences, or whether different elements in presequences determine their specific properties. However, some presequences were found to be more efficient than others [34,35]. 'Strong' presequences may

have been developed for proteins that are difficult to import owing to their hydrophobicity or folding characteristics [36,37]. Alternatively, particularly important proteins might have more efficient presequences to prioritize them under competing import conditions [34,38].

In this study, we systematically compared the features of presequences in the context of standardized fusion proteins. By using in vitro and in vivo assays, that measure mitochondrial protein import efficiency, we found that the 'strong' presequence of Oxa1 lost its advantage as soon as cytosolic binding factors were removed. The cytosolic protein TOMM34, a co-chaperone of the Hsp70-Hsp90 system of mammalian cells [39,40], specifically bound to the Oxa1 presequence, but not to the 'weak' presequence of Atp5. Depletion of TOMM34 or deletion of Tom70 reduced the import efficiency of the Oxa1 presequence. The yeast co-chaperone Cns1 is structurally related to TOMM34 and Cns1 deficiency depletes many mitochondrial proteins, consistent with a TOMM34-equivalent role in mitochondrial protein import. Our results are compatible with a model according to which 'strong' presequences specifically recruit components of the cytosolic chaperone network to improve mitochondrial targeting in a Tom70-assisted fashion.

## Results

### Presequences differ in length and structural characteristics

Presequences are a characteristic feature of mitochondrial proteins and part of the primary structure of almost all matrix proteins [7,41], serving as targeting signals for mitochondria (Fig 1A). Fig 1B shows the presequences of previously characterized model proteins with the characteristic alternating occurrence of positive and hydrophobic residues that gives rise to the amphipathic nature of presequences [6]. These examples also demonstrate the highly variable lengths of presequences. TargetP prediction [42] assigns high propensity scores for mitochondrial targeting to most of the presequences, but also here a certain heterogeneity is apparent (Fig 1C). We wondered whether the features of the presequences of mitochondrial proteins in yeast allow it to define distinct subgroups. Since we felt that lengths and prediction scores might be insufficient, we followed an unbiased machine learning approach. To this end, we developed a novel workflow that mimics in vitro experiments (S1A Fig). Our in silico analysis starts with the engineering of proteins by adding presequences to mouse dihydrofolate reductase (DHFR). These engineered proteins are then embedded in a multidimensional space using a protein language model. We convert each protein sequence into a 768-element vector that captures its properties (see Materials and methods for details). This high-dimensional data is subsequently reduced to a two-dimensional scatter plot using Uniform Manifold Approximation and Projection (UMAP). A set of presequences in an initial set of previously characterized mitochondrial precursor proteins separated into two distinct clusters in the UMAP scatter plot (Fig 1D). These clusters closely resembled our initial in vitro findings, prompting us to extend this in silico approach to all experimentally verified mitochondrial presequences. We identified seven distinct groups (Fig 1E, groups A to G) based on their complex physicochemical characteristics. These groups remained consistent even when sequences from animals, plants, and other fungi (S1 Table) were added to the analysis, indicating that their features are conserved among eukaryotes (S1B Fig). The presequences show a group-specific length distribution, but the categories did not correlate with protein abundance (Fig 1F). Interestingly, the groups are populated by proteins of distinct physiological and biochemical function and protein abundance (Figs 1G and S2 Fig). For example, many enzymes of the respiratory chain and the tricarboxylate cycle are found in group A and B, mitoribosomal proteins (MRPs) are enriched in group C, and uncharacterized proteins are enriched in group G. The uneven distribution of the different categories of mitochondrial proteins within these groups suggests that presequences contain structural information that is relevant in context of the physiological or structural features of the mature proteins which they target into mitochondria.

To test whether the seven groups of presequences show consistent features in respect to their import properties, we used published datasets of yeast mutants in which components of the import machinery were compromised (S1C Fig, S2 Table). This showed that individual groups (such as A, B, C, and F) were strongly affected by specific conditions,

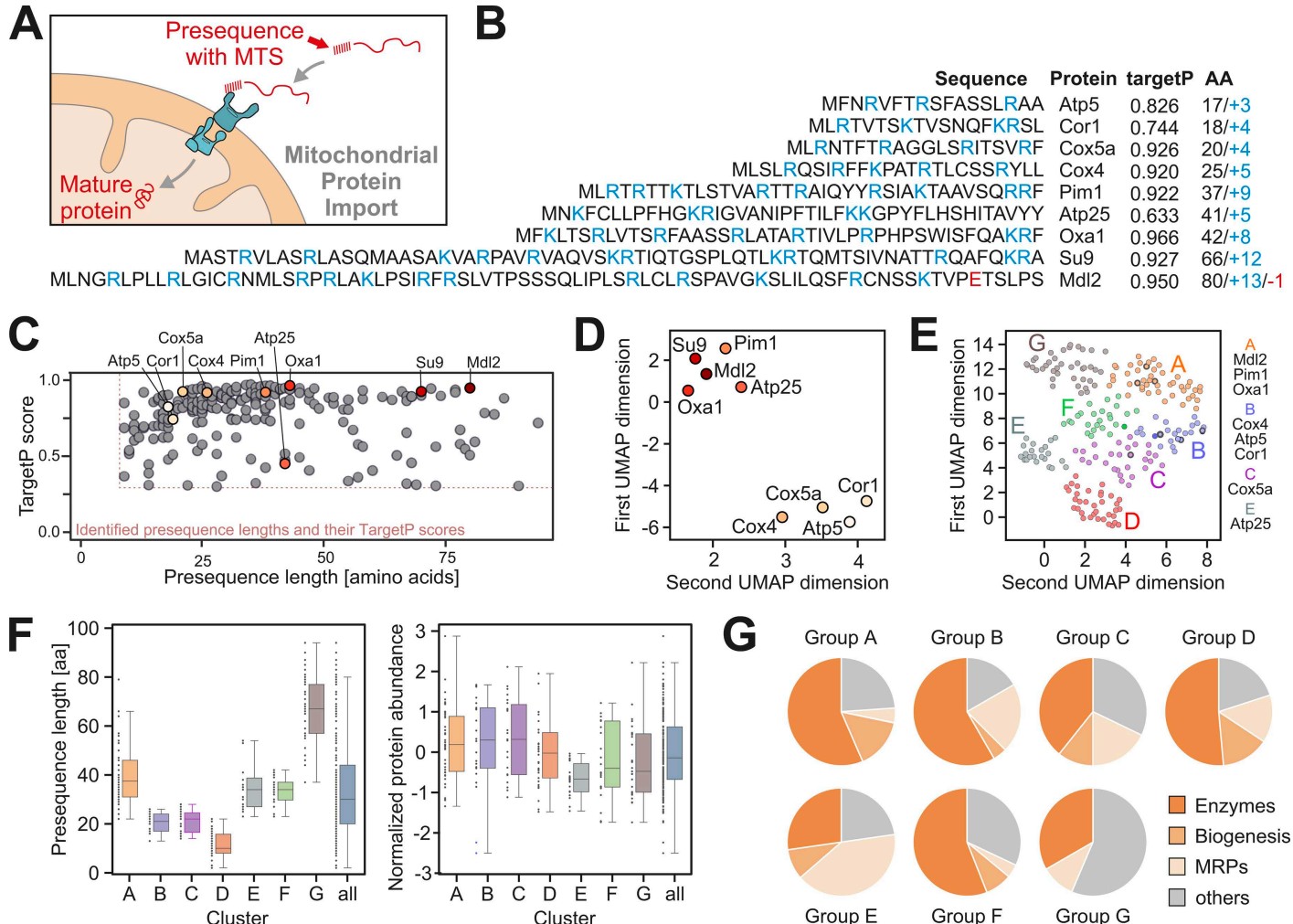

**Fig 1. Properties of mitochondrial presequences. (A)** Schematic representation of mitochondrial protein import. **(B)** Sequences of previously characterized precursor proteins. Shown are the presequences of *Saccharomyces cerevisiae* proteins as well as Su9, the subunit 9 of the ATPase of *Neurospora crassa*, a widely used standard presequence in the field. The sequences up to the most C-terminal cleavage sites are shown; maturation sites had been determined experimentally before [7]. Positively and negatively charged residues are shown in blue and red, respectively. The targetP 1.0 values of these proteins [42] and the number of amino acids (AA) and charged residues of the presequences are indicated. **(C)** TargetP 1.0 scores and lengths distribution of presequences from mitochondrial proteins of *S. cerevisiae* for which the processing sites had been determined [7]. Shown are only proteins with TargetP scores ≥0.3 and lengths >7 residues. Positions of the model presequences used in this study are indicated. The color intensity correlates with length. **(D)** Physiochemical embedding of nine selected MTS (Su9, Pim1, Mdl2, Oxa1, Atp25, Cox4, Cor1, Cox5a, Atp5) reveals a separation into two highly distinct clusters. This embedding, based solely on sequence information transformed by language model features representing physiochemical properties, shows a strong correlation with the import efficiency of the respective proteins. **(E)** Spectral clustering of the physiochemical embedding of all presequences from the *S. cerevisiae* dataset reveals seven distinct clusters, indicating relevant variation among different groups of MTS. Cluster membership aligns with the import efficiency of the respective proteins as follows: group A (Mdl2, Oxa1, Pim1), group B (Cox4, Cor1, Atp5), group C (Cox5a), and group E (Atp25). **(F)** Presequence lengths [7] and protein abundance in galactose-grown yeast cells [1] in the different groups. **(G)** Distribution of different functional categories of proteins in the different groups. See S1B Fig for more information. MRPs, mitoribosomal proteins. The data underlying the graphs shown in the figure can be found in S1 Data.

indicating that the type of presequence correlates with the dependence of the proteins on factors such as Tom70 or on the respiratory competence of the mitochondria. This is consistent with the assumption that the features of the presequences provide specific properties to the respective mitochondrial protein.

## Presequences dictate the import efficiency of precursors into isolated mitochondria

Next, we tested the targeting efficiencies of different presequences experimentally using an established in vitro import assay with isolated yeast mitochondria [43,44]. To this end, we generated a test set of nine model proteins carrying the presequences shown in Fig 1B from the start methionine to 10 residues downstream of the MPP cleavage site to ensure reliable intramitochondrial processing. We chose three representatives each of group A (Mdl2, Oxa1, Pim1) and B (Atp5, Cor1, Cox4), and one for C (Cox5a) and E (Atp25), as well as the presequence of the well-characterized model protein subunit 9 of the ATP synthase of *Neurospora crassa* (Su9). These presequences were fused to DHFR whose import properties are well characterized and can be well detected by western blotting and autoradiography [14]. Thus, these nine proteins differed only in the presequence but carried the identical mature protein. For control, we also used DHFR that did not carry an MTS.

These fusion proteins were synthesized in the presence of $^{35}$S-methionine in reticulocyte lysate and incubated for 5 or 15 min with isolated yeast mitochondria. Non-imported protein was removed by the addition of proteinase K (PK) and samples were analyzed by SDS-PAGE and autoradiography (Figs 2A and S3A). DHFR lacking an MTS did not result in a protease-protected protein and thus was not imported into mitochondria. In contrast, all presequence-containing fusion proteins were processed to a faster migrating, protease-protected mature protein (representing the DHFR domain); however, the import efficiency varied considerably (Fig 2B). Apparently, in this in vitro assay, some presequences were much 'stronger' than others; the presequences of Oxa1, Mdl2, Pim1, and Su9 were extremely efficient. Interestingly, all these highly import-efficient examples belonged to the group A of the in silico analysis, indicating that presequences of this group carry structural determinants that endow them with an increased import competence.

Protein import into mitochondria is driven by the membrane potential across the inner membrane and ATP hydrolysis in the matrix (Fig 2C). Mutants that lack cytochrome oxidase (Δ*cox18*) or lack the proton pumping $F_o$ segment of the mitochondrial ATP synthase (Δ*atp6*) have impaired levels of the membrane potential or ATP synthesis, respectively, and therefore cannot grow on non-fermentable carbon sources (Fig 2D). The import efficiency into these compromised mitochondria was generally reduced (Figs 2E, 2F, S3B, and S3C). However, the 'strong' presequences remained the strongest, and all proteins were affected to some degree by lower energization. We therefore conclude that the differences are not due to different intramitochondrial requirements but presumably by a step upstream, i.e., by the targeting to and interaction with mitochondrial surface receptors.

## Development of a novel GFP-based assay to monitor mitochondrial import efficiency in living yeast cells

Next, we wondered whether the differences in import efficiency are owing to the specific conditions of the in vitro assay or whether they are also of physiological relevance in vivo. Several assays to monitor the import efficiency in vivo had been developed in the past such as pulse-chase assays [45], dynamic SILAC labeling [46,47], protein colocalization by fluorescence microscopy [34], split luciferase assembly in the matrix [48] and the (mis)targeting of Ura3-fused mitochondrial precursor proteins [23]. None of these assays seemed perfect here as they either are not quantitative or not suited for the comparison of the different reporter proteins. We therefore set out to develop a novel assay relying on fluorescence de-quenching [49], which we named IQ-Compete for Import and de-Quenching Competition assay (Figs 3A and S4A). For this method, we generated a continuously expressed GFP protein fused to a quencher sequence, which abolishes fluorescence. We inserted a cleavage site for a yeast-optimized version of the Tobacco Etch virus protease (uTEV) [50] between these two parts. Since this fusion protein was partially de-quenched by insertion of the quencher domain into the ER membrane (S4B Fig), we added a degron to its C-terminus to rapidly deplete the protein and wipe out all background

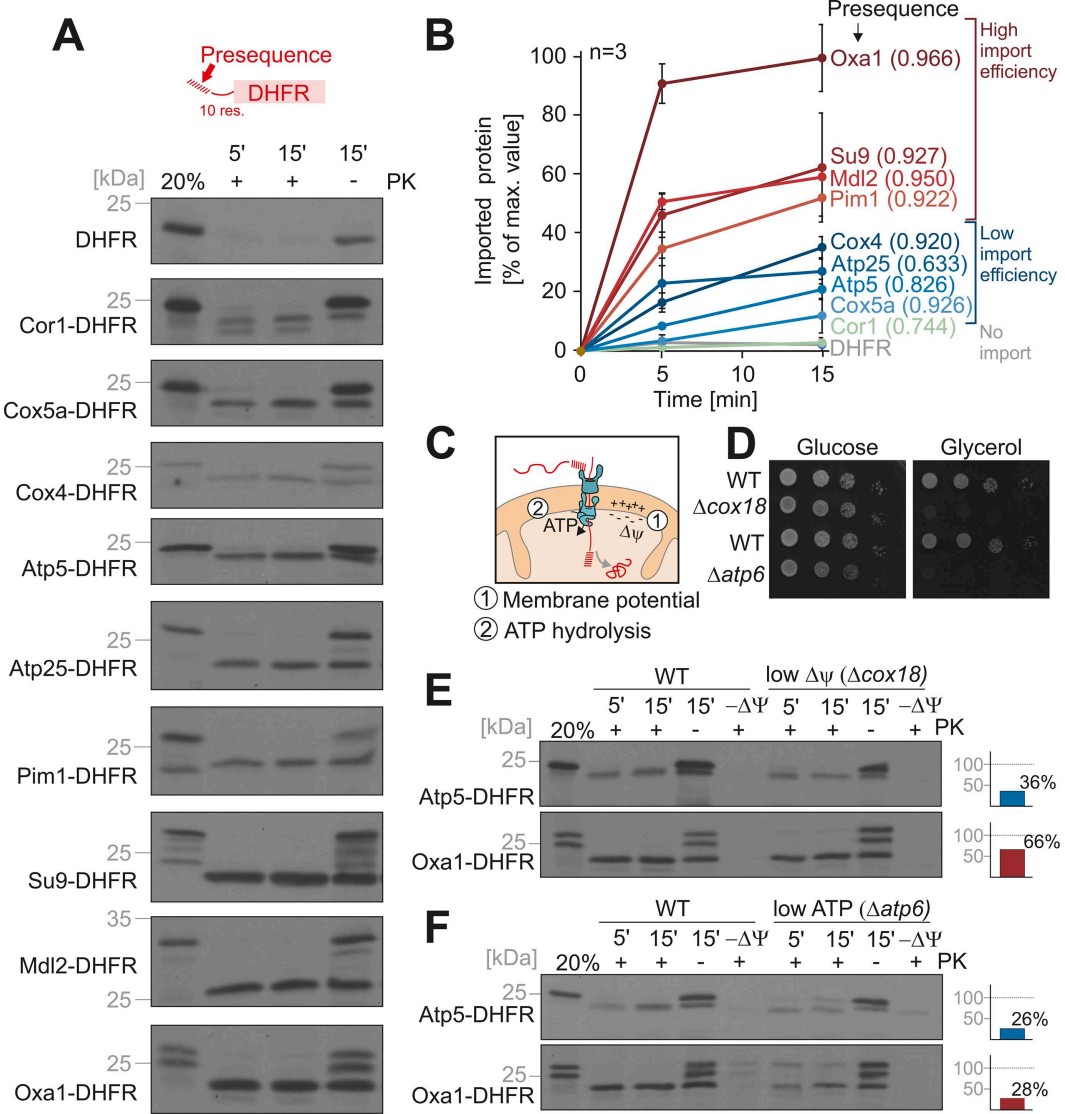

**Fig 2. Presequences determine the import efficiency into isolated mitochondria. (A)** Fusion proteins consisting of the indicated presequences up to 10 residues after the maturation site were fused to mouse dihydrofolate reductase (DHFR). The proteins were synthesized in reticulocyte lysate in the presence of $^{35}$S-methonine and incubated with purified wild type (WT) mitochondria for 5 or 15 min in the presence of 2 mM ATP and 2 mM NADH. Mitochondria were 10-fold diluted into ice-cold sorbitol buffer and further incubated on ice in the presence or absence of 100 µg/ml proteinase K (PK). Mitochondria were isolated by centrifugation and dissolved in sample buffer. 20% of the reticulocyte lysate used per import time point was loaded for comparison. All samples were resolved by SDS-PAGE and visualized by autoradiography. **(B)** The signals of the PK-resistant proteins (+PK) relative to the mitochondria-bound precursor protein (right lanes, −PK) were quantified from three biological replicates (see S2A Fig). **(C)** Schematic representation of the energy sources that drive import into the mitochondrial matrix. **(D)** Cells of WT or the indicated mutants were grown to log phase in galactose medium before 10-fold serial dilutions were dropped on plates containing glucose or glycerol as carbon sources. Mutants lacking Cox18 or Atp6 are unable to grow on non-fermentable carbon sources. **(E, F)** Import reactions were carried out into mitochondria of the indicated strains as described for A. For E, ATP and for F, NADH were omitted from the import buffer. Import efficiencies were calculated as described for B from three biological replicates. Shown are the import efficiencies for 15 min import into mitochondria of the mutant relative to those into WT mitochondria. The data underlying the graphs shown in the figure can be found in S1 Data.

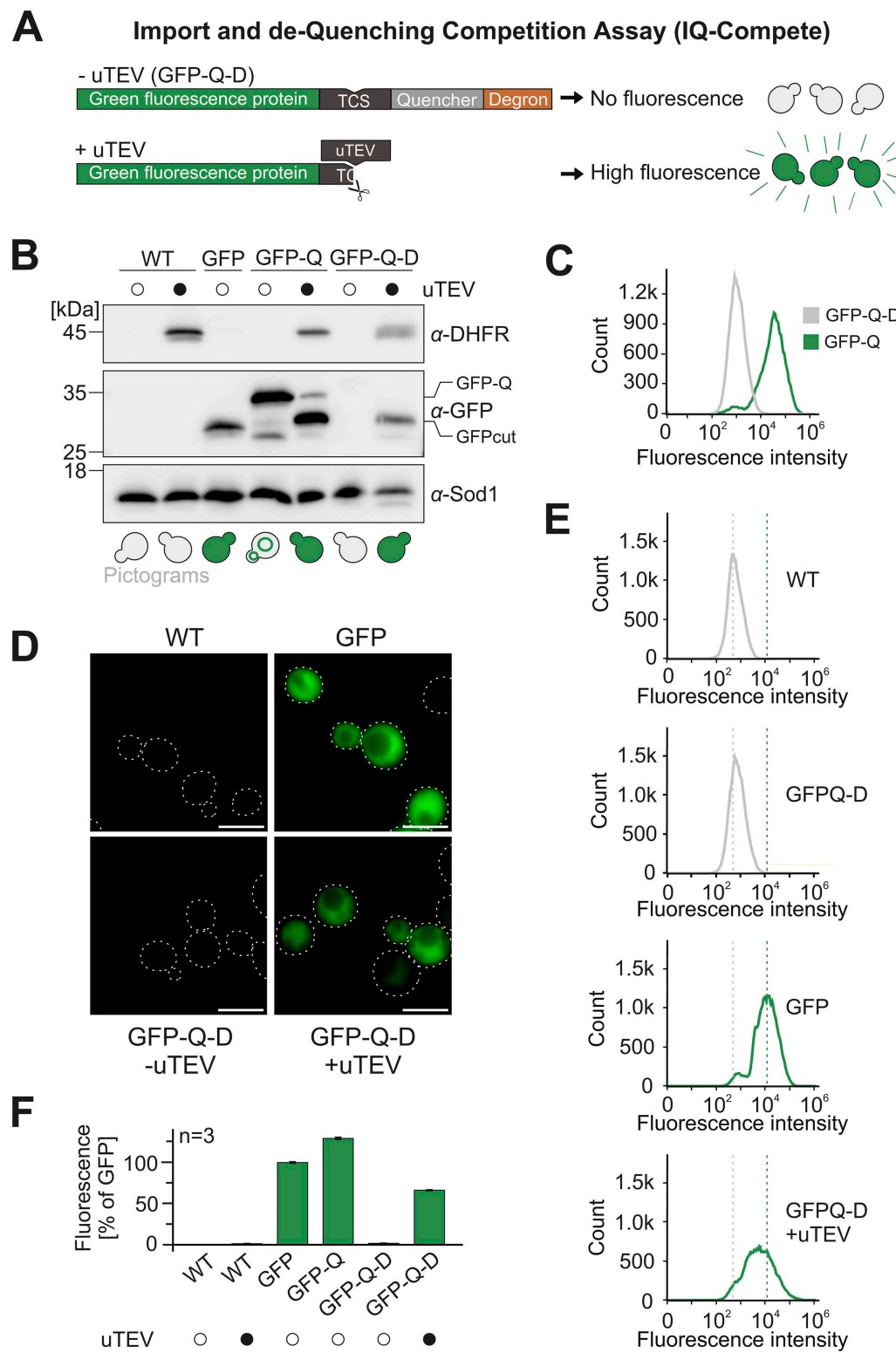

**Fig 3. Fluorescence de-quenching as an efficient strategy to monitor protein levels in the cytosol. (A)** Schematic representation of the reporter construct used for the IQ-Compete assay. TCS, TEV cleavage site. **(B)** WT cells were transformed with plasmids for the expression of uTEV-DHFR as well as of GFP, GFP-quencher (GFP-Q) or GFP-quencher-degron (GFP-Q-D). The cells were grown to mid-log phase in lactate medium and analyzed

by western blotting. The pictograms schematically show the distribution of the GFP signal in these cells. **(C)** WT cells expressing GFP-Q and GFP-Q-D were grown to mid-log phase and the fluorescence intensity was analyzed by flow cytometry. 100,000 cells were analyzed for each sample. **(D)** The indicated strains were analyzed by fluorescence microscopy. All images are shown with identical settings. Scale bar 5 μm. **(E)** Fluorescence intensity of WT cells expressing the indicated constructs. **(F)** Fluorescence intensities were measured and quantified as shown for C in three independent samples. Shown are mean values and standard deviations.

fluorescence (Figs 3B, 3C, and S4C). Thus, cells expressing this GFP-quencher-degron (GFP-Q-D) reporter remained non-fluorescent in the absence of the uTEV protease but showed a strong fluorescence signal once uTEV was expressed in the cytosol. This signal was detectable by fluorescence microscopy (Fig 3D), by measuring fluorescence in a multiplate fluorescence spectrometer (S4D Fig), and by flow cytometry (Fig 3E and 3F).

In a next step, we fused the uTEV protease to the C-terminus of the model proteins consisting of the different presequences and DHFR (Fig 4A). Rapid and efficient import should deplete these precursors and thus the uTEV protease from the cytosol, so that the GFP signal remains quenched. However, slow or incomplete import should result in the cleavage of the reporter and hence in fluorescence. We integrated the GFP-Q-D reporter and the uTEV protease into the yeast genome, which strongly reduced cell-to-cell heterogeneity and resulted in very consistent and reliable fluorescent signals.

First, we tested the levels of the GFP-containing fragment in cell extracts by western blotting (Figs 4B and S4E). This fragment was well detectable upon expression of the uTEV without an MTS in the cytosol (Fig 4B, left lane), but absent without uTEV expression (Fig 4B, right lane). Expression of the presequence-uTEV fusion proteins resulted in high to moderate GFP levels for the samples containing presequences that were less efficient in the in vitro import assay, whereas cells expressing the 'strong' presequences of Oxa1, Su9, and Mdl2 produced no GFP fragment, indicating that these presequences efficiently target the uTEV enzyme into mitochondria. This was even more apparent when the fluorescence of the cells was measured by microscopy (S4F Fig) and, in a quantitative way, by flow cytometry (Fig 4C and 4D). The rather narrow peaks demonstrate the highly consistent fluorescence in the different strains, indicative for a low cell-to-cell heterogeneity. Overall, the fluorescence signals of the IQ-Compete assay thereby strongly correlated with the result of the in vitro import assay (Fig 4E), confirming that the group A presequences provide an increased import efficiency which is also relevant in vivo.

## The 'strong' group A presequence of Oxa1 recruits a cytosolic factor

What is the molecular mechanism that gives priority to group A presequences? Since the differences were also found in the in vitro import assay, we wondered whether a cytosolic factor is critical for the increased import behavior of the Oxa1-DHFR fusion protein. Therefore, we precipitated the radiolabeled precursors with ammonium sulfate and denatured the proteins in 8 M urea. It was shown before that the denaturation of DHFR fusion proteins often increases their import efficiency [51]. Indeed, denaturation of Atp5-DHFR improved its import in vitro; in contrast, denaturation of the Oxa1-DHFR and Su9-DHFR precursor reduced their import efficiency (Figs 5A, 5B, S5A, and S5B). Apparently, the strong difference in the import efficiency of both precursors was abolished by their urea denaturation, suggesting that a cytosolic import-stimulating factor associates with the Oxa1-DHFR precursor.

For the import assays, the reticulocyte extract with the precursor proteins is diluted 100-fold into the import buffer. We tested whether the incubation time of this diluted precursor prior to the addition of mitochondria affects the import efficiency and found that longer preincubations do not affect the import of Atp5-DHFR, but strongly affect that of Oxa1-DHFR (Fig 5C and 5D), suggesting that the cytosolic factor might be released upon prolonged incubation in the buffer. The loss of import efficiency correlated with the aggregation of Oxa1-DHFR protein which was largely found in a high-speed centrifugation pellet after prolonged incubation (Fig 5E and 5F); high concentrations of bovine serum albumin (30 mg/ml BSA) reduced Oxa1-DHFR aggregation. The propensity of mitochondrial presequences to promote the formation of aggregates had been reported before from in vitro [52] and in vivo [29,53] studies. High protein concentrations can maintain

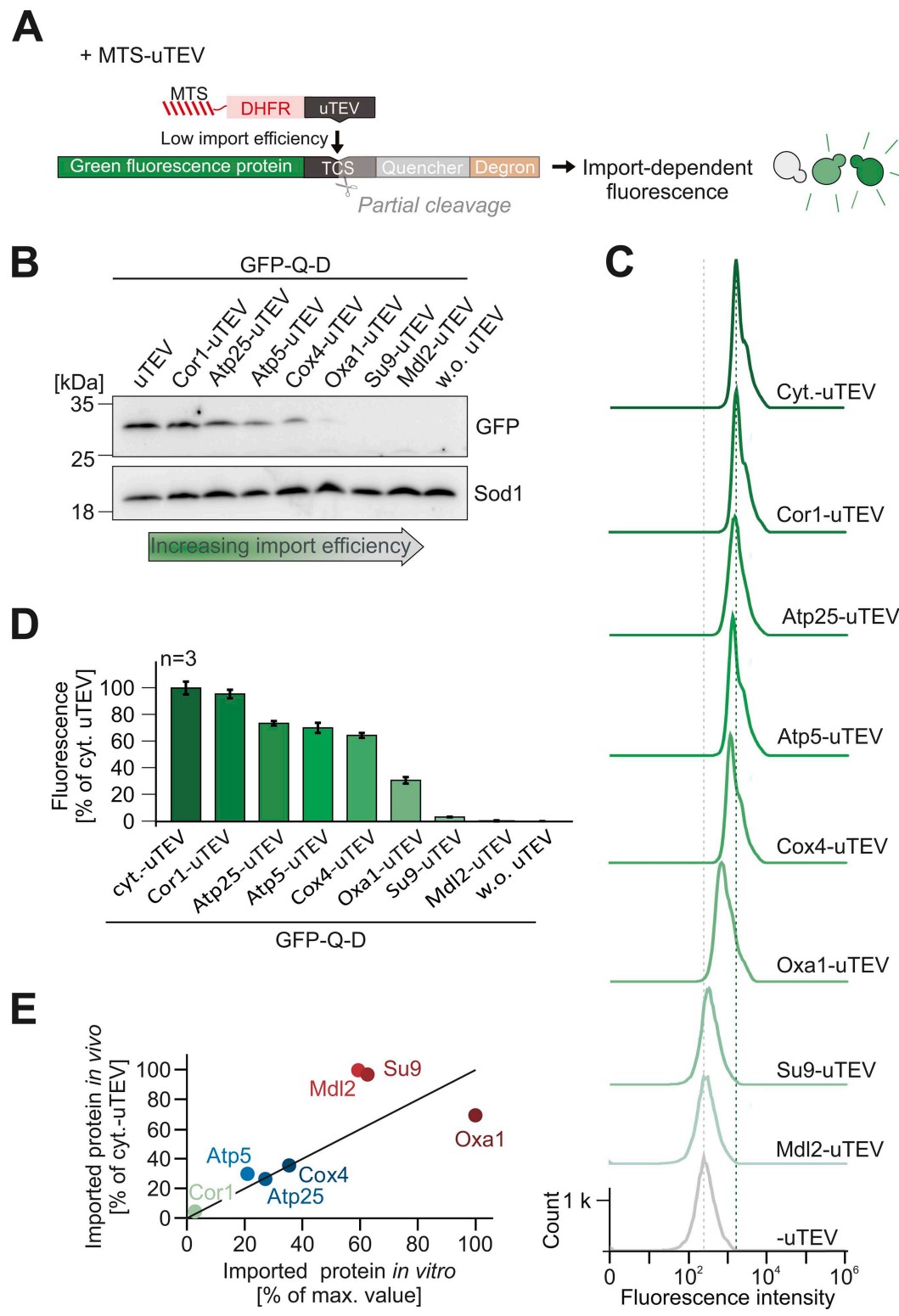

**Fig 4. The in vivo import assay confirms the high import efficiency of group A presequences. (A)** Fusion proteins, consisting of a presequence, DHFR and uTEV, can be used to monitor their import efficiency in vivo. The levels of these proteins in the cytosol determine the de-quenching of the reporter and thereby the fluorescence in individual yeast cells. **(B)** WT cells expressing the GFP-quencher-degron reporter and the indicated uTEV

constructs were grown to mid-log phase and analyzed by western blotting. The GFP signal indicates the amount of the cleaved GFP-Q-D protein. **(C, D)** Fluorescence intensities were measured in WT cells expressing GFP-Q-D plus the indicated proteins. Quantifications show mean values and standard deviations of three independent replicates. **(E)** Correlation of the import efficiencies observed in vitro (Fig 2B, 15 min) and in vivo (Fig 4D). Group A proteins are shown in red. The data underlying the graphs shown in the figure can be found in S1 Data.

precursors in a soluble and import-competent state (Fig 5G and 5H). Moreover, the import efficiency of Oxa1-DHFR dropped when lower amounts of mitochondria was used in the import assay (Fig 5I and 5J), potentially because lower concentrations of mitochondria lead to a prolonged targeting of the precursors in the import buffer [54].

Chaperones of the Hsp70 and Hsp90 family as well as their co-chaperones such as J domain proteins are known to stimulate the import of some precursors in a Tom70-dependent manner [19,21,22]. We therefore tested the relevance of Tom70, and its homolog Tom71, for the import of Atp5-DHFR and Oxa1-DHFR. Whereas the presence of Tom70/Tom71 was not relevant for the import of Atp5-DHFR, the import of Oxa1-DHFR was considerably less efficient in the absence of Tom70 and its homolog Tom71 (Fig 5K and 5L), suggesting that the efficient nature of the 'strong' presequence is at least in part due to the interaction of a cytosolic protein with Tom70 (Fig 5M).

### The tetratricopeptide repeat (TPR) protein TOMM34 binds to the 'strong' presequence of Oxa1

To identify cytosolic factors that bind to the different model proteins, we synthesized Atp5-DHFR, Oxa1-DHFR, Su9-DHFR, and DHFR in reticulocyte lysate (without radiolabeling), isolated these proteins by immunoprecipitation using DHFR-specific antibodies and identified the full spectrum of bound proteins in the precipitate unambiguously by mass spectrometry. Principal component analysis showed that the presence of presequences caused a characteristic footprint on the interactome (Fig 6A). The Hsp70 protein 8 (homologous to the human HSP A8 protein) was strongly enriched in the pull-downs with Atp5-DHFR, Oxa1-DHFR, and Su9-DHFR, but not with DHFR (Figs 6B, S5C, and S5D and S3 Table). Thus, this cytosolic Hsp70 chaperone apparently serves as general presequence-binding factor in the reticulocyte lysate, consistent with a recent study [27]. The J domain protein DNAJC13 was also consistently found with presequences, although at lower levels (S5D Fig). Ubiquilin, which was reported to bind to the transmembrane domains of cytosolic precursors of inner membrane proteins, was not pulled down. This was expected as the DHFR fusion proteins lacked hydrophobic sequences that could serve as ubiquilin-binding sites [58].

Next, we tested which proteins of the reticulocyte lysate specifically bound to the 'strong' presequence of Oxa1 (Fig 6C). The protein TOMM34 was one of the factors that were significantly enriched with Oxa1-DHFR but not with Atp5-DHFR. TOMM34 has been identified before as a factor in the human cytosol that associates with the mitochondrial surface and stimulates protein import into mitochondria [40]. TOMM34 contains six TPRs (Fig 6D), a pattern that is frequent in many co-chaperones of the Hsp70/Hsp90 system as well as in Tom70 [59]. TOMM34 is found in animals but not in fungi, however, its sequence is similar to that the Cns1 protein of fungi (Fig 6E) which, like TOMM34, serves as Hsp90 co-chaperone [60].

To identify the TOMM34 binding site in the Oxa1 presequence, we used a peptide spot assay [61] which has been successfully used to determine the binding sites of TOM receptors in presequences [62]. To this end, we coupled peptides of 20 residues each onto a cellulose membrane that covered the presequences of Oxa1, Atp5, and other precursor proteins (using a step size of three residues). We purified recombinant TOMM34 from *Escherichia coli* (S5E Fig) and incubated it with the membrane. After intensive washing, the bound TOMM34 protein was detected by western blotting (Figs 6F and S5F). A peptide in the C-terminal part of the Oxa1 presequence showed high affinity for TOMM34, confirming the Oxa1-TOMM34 interaction. The highly variable binding intensities to the different regions of presequences suggest that TOMM34, rather than binding to MTS in general, associates with a subgroup of presequences via specific binding sites, potentially to promote the binding to Hsp90 and Tom70 (S5G Fig).

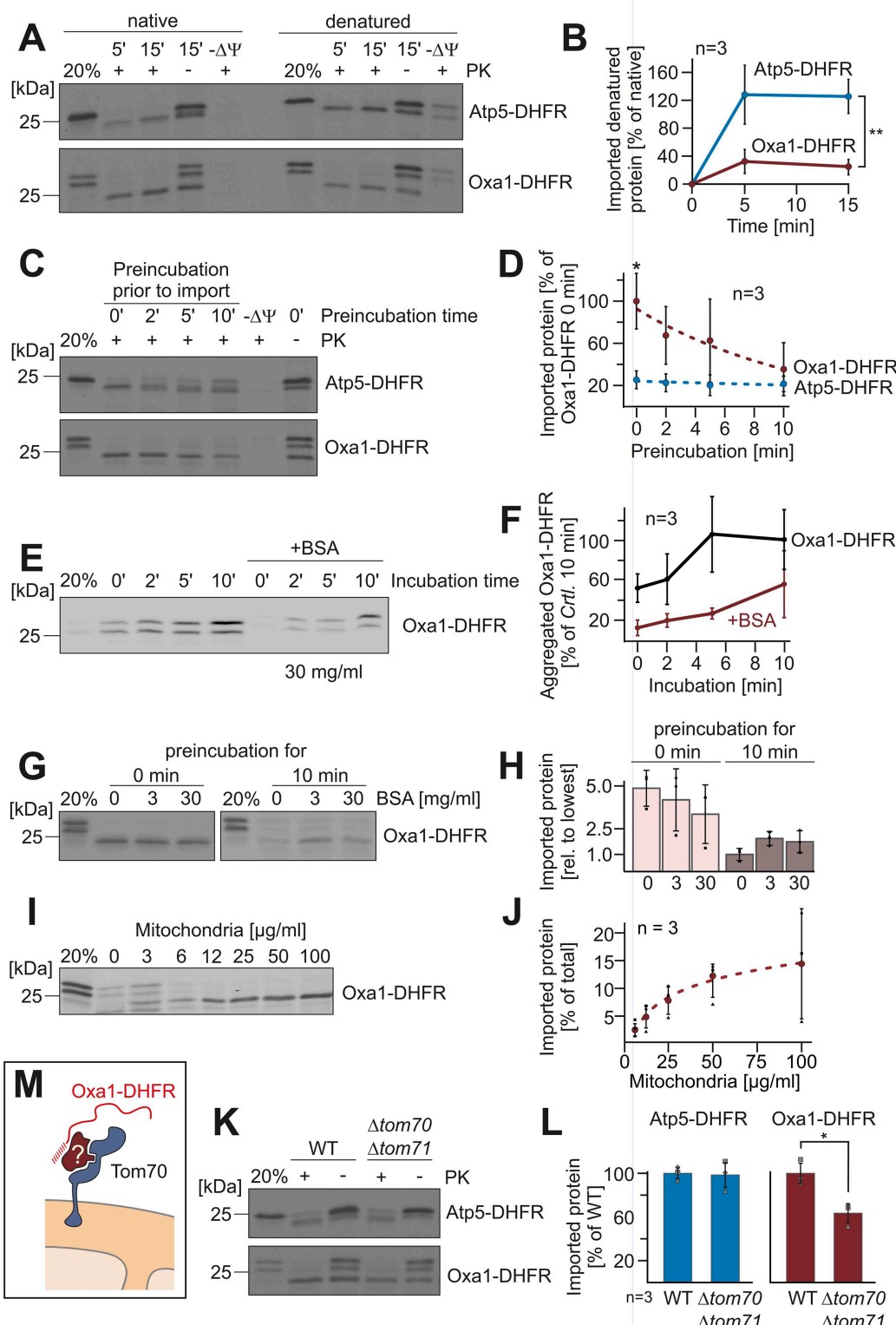

Fig 5. A cytosolic factor increases the import efficiency of the 'strong' presequence of Oxa1. (A) The radiolabeled precursor proteins were precipitated and dissolved in 8 M urea, 100 mM DTT, and 20 mM Tris pH 7.4. Imports were performed as described for Fig 2A using either radiolabeled proteins in reticulocyte extracts (native) or the urea-treated proteins (denatured). (B) The relative amounts of imported proteins were quantified from three biological replicates. Shown are mean values and standard deviations. Statistical differences were calculated with a Student t test. Statistical

significance was assigned as follows: *p*-value < 0.01 = **. **(C)** Precursor proteins were 100-fold diluted into import buffer containing ATP and NADH and incubated at 30 °C. After different times, the import reaction was started by addition of isolated mitochondria. After 15 min, the import reactions were stopped and analyzed. **(D)** The amounts of imported proteins were quantified and plotted relative to the import efficiency of the non-preincubated Oxa1-DHFR sample. Shown are mean values and standard deviations. Statistical differences were calculated with a Student *t* test. Statistical significance was assigned as follows: *p*-value < 0.1 = *. **(E, F)** Reticulocyte lysate containing the radiolabeled Oxa1-DHFR precursor was diluted 100-fold in import buffer in the absence or presence of 30 mg/ml BSA. After incubation at 30 °C for the indicated times, aggregated proteins were precipitated by centrifugation at 30,000 *g* for 15 min and subjected to SDS-PAGE and autoradiography. The signals of aggregated proteins were quantified from three independent experiments and plotted relative to the signal of Oxa1-DHFR after 10 min incubation without BSA. Shown are mean values and standard deviations. **(G, H)** Radiolabeled Oxa1-DHFR was incubated in the absence or presence of BSA (3 or 30 mg/ml) for 0 or 10 min (preincubation), before mitochondria were added to start the import reaction. After 15 min, non-imported protein was degraded by protease treatment. The amounts of imported protein were quantified from three independent experiments and are shown relative to the imported protein that was imported after preincubation without BSA. Shown are mean values and standard deviations. **(I, J)** Radiolabeled Oxa1-DHFR was incubated with different amounts of isolated mitochondria for 15 min at 30 °C. Non-imported protein was removed by treatment with proteinase K. The amounts of imported protein were quantified from three independent experiments. Shown are mean values and standard deviations. **(K)** The indicated precursor proteins were imported into mitochondria of wild type or Δ*tom70* Δ*tom71* double deletion mutants for 15 min and further treated as described for Fig 2A. **(L)** The import efficiencies were quantified and are shown in relation to that of wild-type cells. Shown are mean values and standard deviations of three independent replicates. Statistical difference was calculated with a Student *t* test. Statistical significance was assigned as follows: *p*-value < 0.1 = *. **(M)** Our results suggest that a cytosolic factor that is present in the reticulocyte lysate and in the yeast cytosol facilitates mitochondrial import in a Tom70-dependent manner. The data underlying the graphs shown in the figure can be found in S1 Data.

## TOMM34 binding promotes the import of Oxa1-DHFR

Next, we asked whether the TOMM34 binding to the Oxa1 presequences is crucial for its high import efficiency. Therefore, we first tested whether TOMM34 binding can suppress the aggregation of Oxa1-DHFR (Fig 7A and 7B). To this end, we diluted the Oxa1-DHFR-containing reticulocyte lysate 10-fold in the presence or absence of 3 mg/ml TOMM34, incubated the solution for different times and isolated the aggregated Oxa1-DHFR by centrifugation. We thereby observed that TOMM34 can strongly suppress the aggregation of Oxa1-DHFR. Moreover, TOMM34 strongly promoted the import of pre-diluted Oxa1-DHFR into mitochondria whereas BSA had no significant effect on the import efficiency (Fig 7C–7E). Thus, TOMM34 not only can prevent precursor proteins from aggregation but also facilitates their import into mitochondria. The stimulating effect of TOMM34 on the import of Oxa1-DHFR was less pronounced if Tom70-deficient mitochondria were used (Fig 7F).

To test whether TOMM34 is relevant in the context of the in vitro import experiments, we used TOMM34 antibodies and added them to radiolabeled Atp5-DHFR and Oxa1-DHFR prior to the import reaction. Pretreatment with TOMM34-specific antibodies, but not with Sod1-specific antibodies that we used for control, strongly reduced the import efficiency of Oxa1-DHFR but not of Atp5-DHFR (Fig 7G and 7H). The physiological relevance of the 'strong' presequence of Oxa1 is also supported by the observation that an Oxa1 version with the 'weaker' Atp5 presequence does not fully complement a temperature-sensitive *oxa1* mutant (Fig 7I).

## In yeast, the co-chaperone Cns1 facilitates mitochondrial protein biogenesis

TOMM34 is not present in yeast, but the co-chaperone Cns1 contains a similar Hsp70/Hsp90-binding domain formed by three TPR domains (Fig 8A). To test whether the essential protein Cns1 is relevant for mitochondrial biogenesis, we used a mutant in which the Cns1 expression can be controlled by doxycycline [60]. The depletion of Cns1 impairs growth on non-fermentable carbon sources indicative for a role of Cns1 in mitochondrial biogenesis (Fig 8B). Western blots also showed strongly diminished levels of the mitochondrial proteins Oxa1 and Mam33, particularly upon growth on glucose and galactose (Fig 6C and 6D). For a more comprehensive analysis, we repressed the expression of Cns1 for 30 h and measured the cellular proteome by quantitative mass spectrometry (S6A Fig). Depletion of Cns1 caused a strong reduction of mitochondrial proteins (Figs 8E and S6B and S4 Table). According to GO term analysis, the constituents of the mitochondrial ribosome were the most affected protein group upon Cns1 depletion (Fig 8F and S5 Table). Moreover,

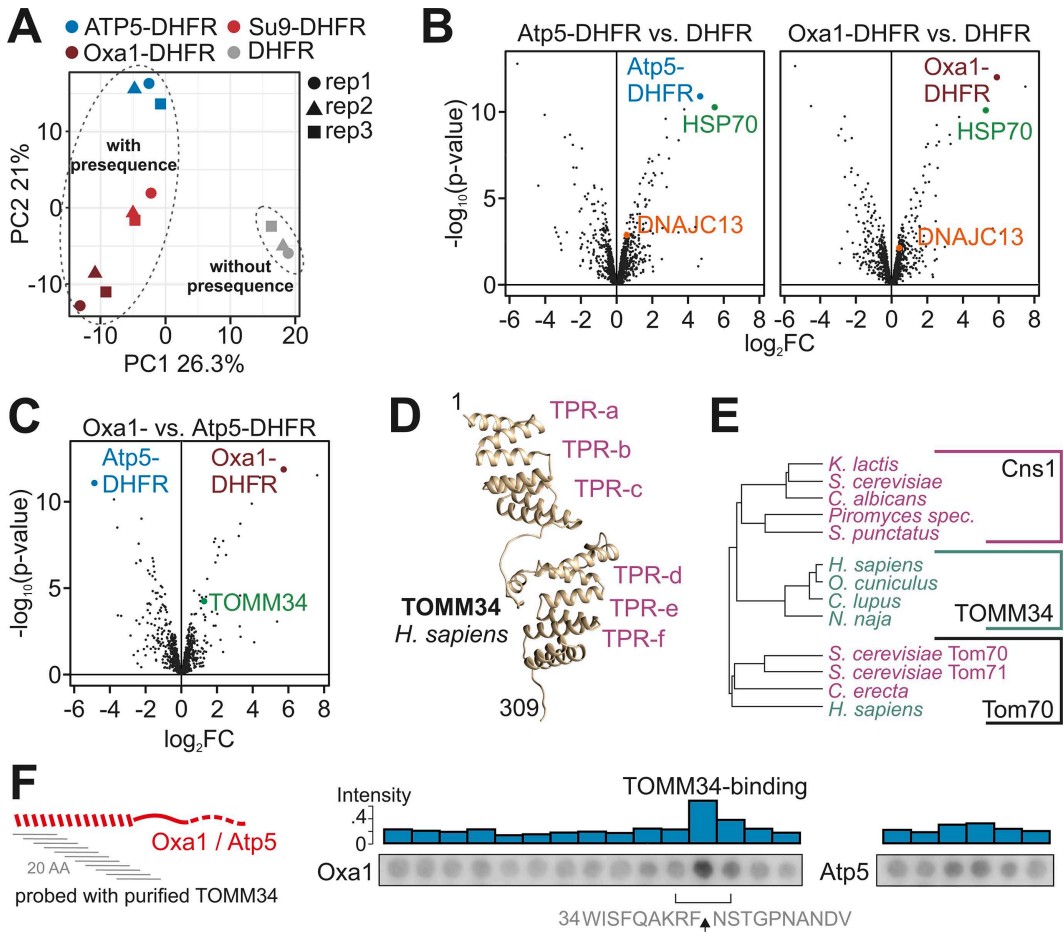

**Fig 6. Presequences recruit specific cytosolic chaperones. (A)** The indicated DHFR fusion proteins were expressed in reticulocyte lysate and purified under mild non-denaturing conditions using DHFR-specific antibodies. Bound proteins were identified by mass spectrometry in three independent replicates. The principal component (PC) analysis shows the similarity of the protein spectrum identified in the different samples, rep, replicate. **(B, C)** Volcano plots comparing the relative enrichments in the immunoprecipitates indicated. HSP70 refers to rabbit Heat shock cognate 71 kDa protein, the homolog of human HSP70 A8; $\log_2$FC, $\log_2$-fold change. The data underlying the graphs shown in the figure can be found in S3 Table. **(D)** The alphafold structure of the human TOMM34 protein was modeled and visualized with the Chimera software [55]. Positions of the six TPR hairpins are indicated. **(E)** Phylogenetic tree showing the relationship of Cns1, TOMM34 and Tom70 sequences. The sequences were aligned, and trees were calculated using the Muscle Multiple Sequence Alignment tool [56,57] using the following sequences: Cns1 (*Kluyveromyces lactis* Q6CVR7; *S. cerevisiae* Cns1; *Candida albicans* A0A8H6BZ74; *Spizellomyces punctatus* A0A0L0HPW1), TOMM34 (*Homo sapiens* Q15785; *Oryctolagus cuniculus* G1SIU6; Canis lupus A0A8I3NTN0; Naja naja A0A8C6X204), Tom70 (*S. cerevisiae* Tom70/Tom71, *Coemansia erecta* A0A9W7XVH4; *H. sapiens* O94826). Species names of animals and fungi are shown in green or red, respectively. **(F)** Peptides of 20 residues in length (step size three residues) representing the presequences of Oxa1 and Atp5 were spotted onto cellulose membranes and incubated with purified TOMM34. After washing, the bound TOMM34 was detected by western blotting and quantified. The sequence of the peptide with highest TOMM34 binding affinity is shown and the MPP processing site is indicated by an arrow. The data underlying the graphs shown in the figure can be found in S1 Data.

we found that Cns1 depletion affected the different presequence groups to different degrees and group E was particularly Cns1-dependent (Fig 8G). Since group E contained many MRPs (Fig 1G), this group might be characterized by Cns1-recruiting presequences. Expression of the human TOMM34 protein in yeast did not restore the observed proteome changes and also did not suppress the growth phenotype of the Cns1 depletion mutant (S6C–S6E Fig). Whether Cns1 really is the functional equivalent of TOMM34 in fungi will have to be assessed in more detail in the future.

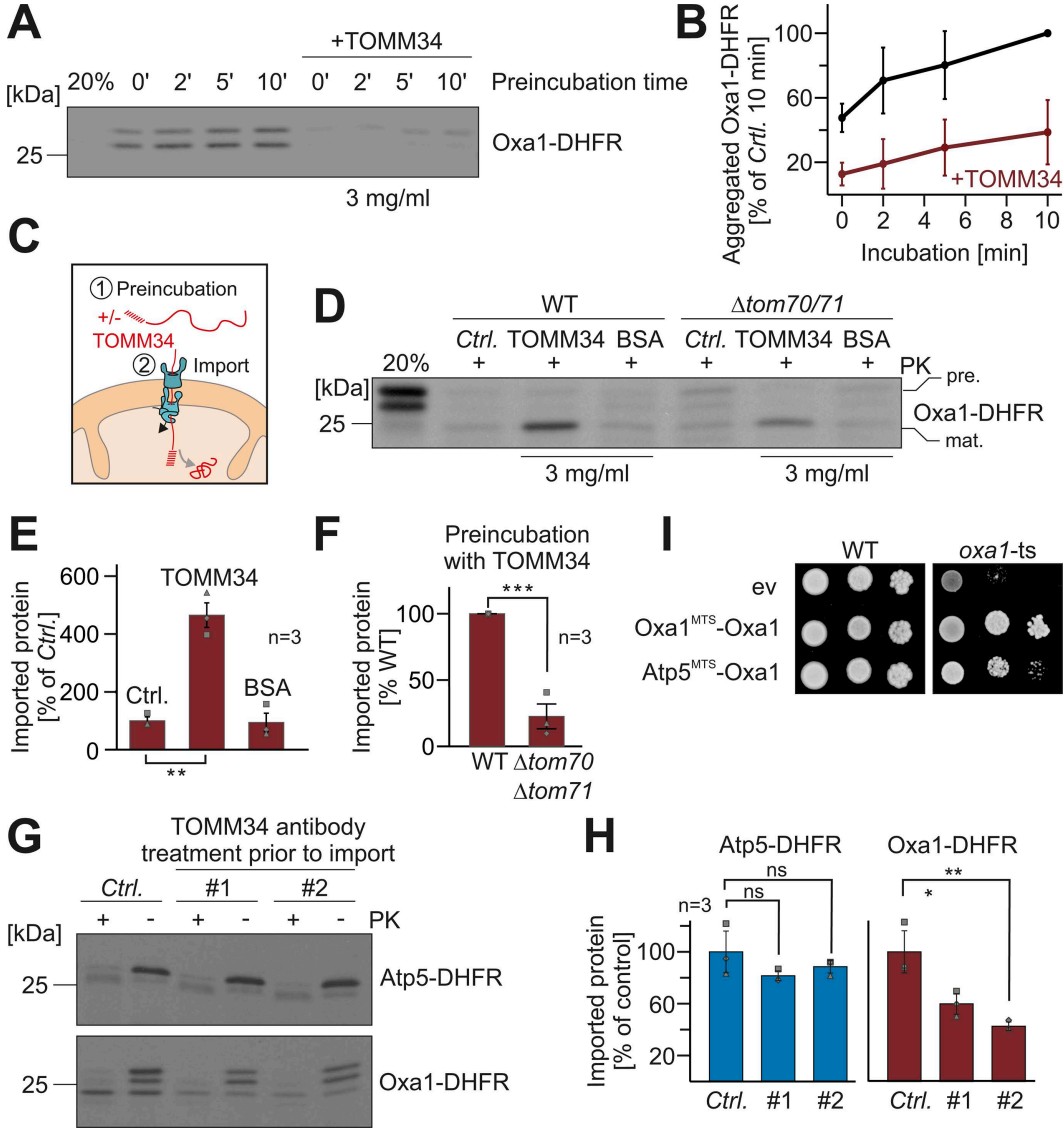

**Fig 7. TOMM34 increases the import competence of Oxa1. (A, B)** Reticulocyte lysate containing the radiolabeled Oxa1-DHFR precursor was diluted 100-fold in import buffer in the absence or presence of 3 mg/ml purified TOMM34. After incubation at 30 °C for the times indicated, aggregated proteins were precipitated by centrifugation at 30,000 *g* and subjected to SDS-PAGE and autoradiography. The signals of aggregated proteins were quantified from three independent experiments and are shown relative to the signal of aggregated Oxa1-DHFR after 10 min incubation without TOMM34. Shown are mean values and standard deviations. **(C–F)** Radiolabeled Oxa1-DHFR was 100-fold diluted in import buffer containing 3 mg/ml BSA or TOMM34 or without added protein (Ctrl.). After incubation for 10 min at 30 °C, mitochondria were added for 15 min. The amount of imported protein was assessed after incubation with proteinase K and quantified from three independent replicates. Statistical difference was calculated with a Student *t* test. Statistical significance was assigned as follows: *p*-value < 0.01 = **. **(F)** The import efficiency in the presence (WT, wild type) and absence of Tom70 (*Δtom70 Δtom71*) after Oxa1-DHFR preincubation with TOMM34 was quantified from three independent experiments. Shown are mean values and standard deviations. Statistical difference was calculated with a Student *t* test. Statistical significance was assigned as follows: *p*-value < 0.001 = ***. **(G)** Reticulocyte lysates with the indicated radiolabeled precursor proteins were incubated with two different antibodies against TOMM34 (#1, #2) or Sod1 for control (Ctrl.) for 15 min at 25 °C. Import experiments were carried out for 15 min as described for Fig 2A. **(H)** Import efficiencies after pretreatment with TOMM34- or control-antibodies were quantified in three independent samples. Two different TOMM34-specific antibodies were used (#1 and #2). Shown are mean values and standard deviations. Statistical difference was calculated with a Student *t* test. Statistical significance was assigned as follows: *p*-value < 0.1 = *; < 0.01 = **. **(I)** Plasmids for the expression of Oxa1 with either the presequences of Atp5 (residues 1–17) or the Oxa1presequence (residues 1–42) or an empty vector (ev) as control were transformed into wild type or a temperature-sensitive *oxa1* mutant (*oxa1*-ts). Cells were grown to mid-log phase in galactose medium. Tenfold serial dilutions were dropped on glycerol-containing plates and incubated for 4 days at 30 °C. The data underlying the graphs shown in the figure can be found in S1 Data.

## Discussion

The targeting properties of mitochondrial presequences were identified 40 years ago [63]. Shortly after their first description, their positively charged, amphipathic helix had been identified as the characteristic feature of mitochondrial targeting sequences [6]. The existence of 'stronger' and 'weaker' presequences was repeatedly described in the literature [34,35,64,65], but a systematic analysis was missing so far. Based on the primary structure of yeast proteins, we were able to distinguish seven different groups of presequences. These seven groups still clustered when sequences from proteins of animals and plants were added. Arguably, the number and composition of such groups depends to some degree on the specific settings of the clustering; nevertheless, it is clear that eukaryotic cells employ different categories of presequences that are defined by their specific structural properties.

Using nine different model proteins, we observed that some presequences were much more potent than others. We predominantly focused on groups A and B as these groups contain many proteins of high relevance for respiration and metabolism. Interestingly, the three 'strongest' presequences (Oxa1, Su9, Mdl2) all belonged to group A. Group A presequences are relatively long and the increased import efficiency of some long presequences was observed before [66–68]. It was shown that longer presequences can contain segments that facilitate the binding to TOM receptors to increase the import efficiency [35,62,67,69,70]. This is consistent with our observation that enhanced import efficiency of Oxa1-DHFR depends on the presence of Tom70.

The different 'strengths' of presequences is not an artifact of the in vitro import assay but rather reflects inherent features which are also relevant in vivo. This was obvious from the results of the novel IQ-Compete assay that we specifically developed for this study. This new tool will certainly be very helpful as it monitors the targeting efficiency of mitochondrial precursors in living yeast cells. In contrast to previously used in vivo methods [23,34,45–48] the IQ-Compete assay allows single-cell level analysis and the very narrow distribution of the fluorescence signals in the flow cytometer experiments clearly documents a very low cell-to-cell variability of this assay once the reporter constructs had been integrated into the yeast genome. Moreover, this assay was quantitative, highly reproducible and showed very consistent results on the basis of different detection methods (by western blotting, fluorescence microscopy, microplate-based spectroscopy, and flow cytometry).

Based on the results of this study, we propose a priority code model (Fig 8H). According to this hypothesis, a subset of mitochondrial precursors employs particularly powerful presequences to recruit cytosolic targeting factors. These factors might support the import in two ways: first, by increasing the targeting efficiency to the mitochondrial surface, and second, by the unfolding of precursor proteins to facilitate their translocation through the mitochondrial protein translocases [20,70,71]. Oxa1 is a rather abundant hydrophobic inner membrane proteins with five transmembrane segments. The very efficient presequence might prevent the misfolding or aggregation of Oxa1 in the cytosol. To further increase the targeting efficiency in vivo, Oxa1 can transiently interact with the ER surface to reach the mitochondrial surface via the ER-SURF pathway [23,24].

The cytosolic TPR protein TOMM34 would be perfectly suited for both processes. TOMM34 can be found as a cytosolic and a mitochondria-bound fraction, even though the binding partner on the outer membrane remains to be discovered [40]. Tom70 and the small TOM subunits were proposed to recruit TOMM34 to enhance mitochondrial protein import. TOMM34 consists of two domains, each consisting of three TPR hairpins loops. Both domains are similar to the dicarboxylate clamp domains of Tom70 and HOP; the N-terminal domain preferentially binds to Hsp70 and the C-terminal to Hsp90 [39,72]. Moreover, TOMM34 was reported to bind to some mitochondrial precursors, particularly of inner membrane proteins. Some studies reported that presequence-containing precursors are TOMM34 substrates [40] whereas others claimed that proteins with presequences are imported independently of TOMM34 [73]. Our observation, that only a subgroup of precursors utilizes the TOMM34-enhanced import mechanism, can explain this discrepancy. Interestingly, a recent study reported that in human cells, cytosolic precursor proteins can recruit Hsp90 via the co-chaperones p23 and Cdc37 [27]. These are non-TRP containing co-chaperones of the Hsp90 system which support the reaction cycle

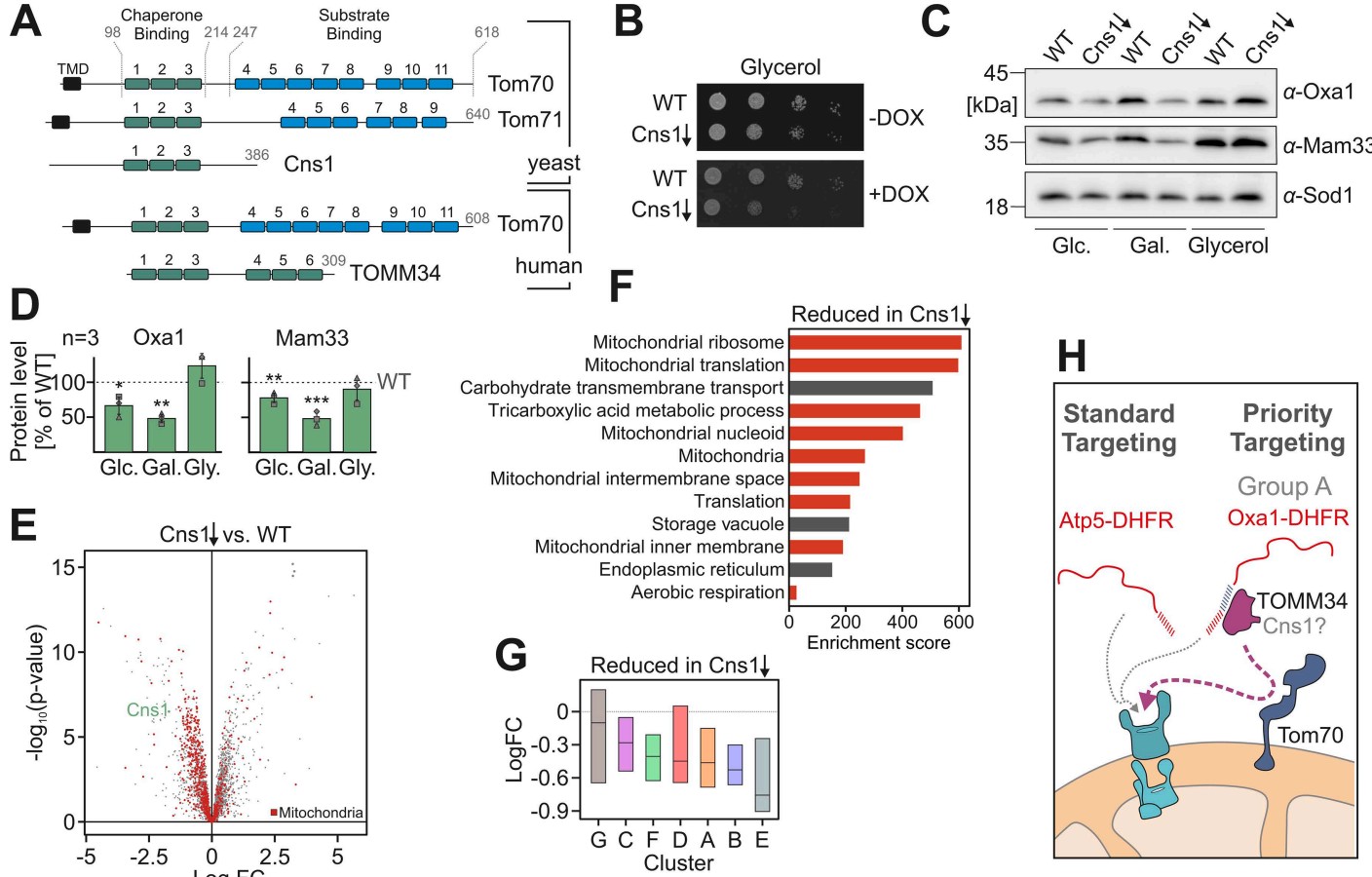

**Fig 8. The TOMM34 homolog Cns1 facilitates mitochondrial protein biogenesis in yeast. (A)** Structural organization of Cns1, TOMM34, and Tom70 proteins. TPR motifs are indicated as boxes. Chaperone and substrate binding regions are indicated in green and blue, respectively. The positions of transmembrane domains (TMD) are indicated. **(B)** The expression of Cns1 was repressed by use of doxycycline-sensitive promoter in front of the *CNS1* gene [60]. Cells were grown to mid-log phase in galactose medium and for 24 h in the presence of 10 µg/ml doxycycline before 10-fold serial dilutions were dropped on the indicated plates. DOX, 10 µg/ml doxycycline (Doxycycline hyclate from Sigma, CAS Number: 24390-14-5). **(C, D)** Cells of the indicated mutants were grown in the media containing glucose (Glc.), galactose (Gal.), or glycerol as carbon source. All samples were treated with doxycycline for 30 h to induce Cns1 depletion. Cell extracts were used for western blotting with the indicated antibodies. Western blot signals were quantified from three independent replicates. 10 µg/ml DOX, doxycycline hyclate. Statistical difference was calculated with a Student *t* test. Statistical significance was assigned as follows: *p*-value < 0.1 = *; < 0.01 = **; < 0.001 = ***. **(E)** WT and Cns1 depletion cells with and without TOMM34 being genomically expressed were grown in galactose-containing medium, treated with doxycycline for 30 h at 30 °C, harvested and subjected to quantitative mass spectrometry. The changes of the whole cell proteomes are plotted; the position of Cns1 is marked and mitochondrial proteins [1] are labeled in red (see also S4 Table). **(F)** Proteins with a log2-fold change < −0.5 and *p*-value < 0.05 in the limma analysis were further analyzed by gene ontology (GO) term enrichment using the GOrilla website (http://cbl-gorilla.cs.technion.ac.il). See S5 Table for details. **(G)** Relative protein levels (log2-fold changes) upon Cns1 depletion were plotted for the representatives of the different presequence groups. **(H)** Model of the enhanced targeting competence of group A presequences: The specific recruitment of targeting factors such as TOMM34 and Cns1 can increase the import efficiency in a Tom70-mediated manner. The data underlying the graphs shown in the figure can be found in S1 Data.

of Hsp90 in steps that are distinct from those carried out by the cytosolic TPR proteins such as TOMM34 [74,75]. Thus, TOMM34, p23, and Cdc37 might cooperate during early steps of mitochondrial protein import.

Fungi lack TOMM34 homologs; however, they contain several Hsp90 co-chaperones with similar structure and function. Cns1, like TOMM34, is a cytosolic TPR protein that contains a dicarboxylate clamp domain for chaperone binding. Similar to TOMM34, it functions as a recruiter protein for Hsp90 to enhance the folding of specific substrate proteins [60]. It also

interacts with Hsp70 and enhances the ATPase activity of Ssa1 [76]. Ssa1 is the predominant precursor-binding chaperone in the yeast cytosol [25]. Thus, just like TOMM34, Cns1 might recruit the Hsp90 system to Hsp70-bound precursors to enhance their targeting and translocation. Cns1-depleted cells were unable to grow on non-fermentative carbon sources, a phenotype characteristic for mitochondrial dysfunction. Even though this is an attractive model, at this point, the role of these Hsp90 co-chaperones in mitochondrial protein biogenesis is still unclear and will have to be characterized in the future in much more depth.

## Materials and methods

### Yeast strains and plasmids

Yeast strains used in this study are based on the wild-type strains W303 (MATα *leu2-3,112 trp1-1 can1-100 ura3-1 ade2-1 his3-11,15*) and BY4741 (MATa his3Δ1 leu2Δ0 met15Δ0 ura3Δ0), except for the Δatp6 mutant and its corresponding wild type which were based on D273-10-B [77] and the Cns1 depletion strain [60]. Yeast strains and plasmids used in this study are described in detail in the S6 and S7 Tables.

For expression of the Oxa1 protein with the presequence of Atp5 (Atp5^MTS-Oxa1), the sequences for the Atp5 residues 1–17 and Oxa1 43–402 were amplified and cloned sequentially into an expression plasmid using a MoClo strategy under control of a constitutive *TEF2* promoter [78]. The Oxa1^MTS-Oxa1 construct was cloned accordingly.

Yeast cells were grown at 30 °C in yeast full medium containing 3% (w/v) yeast extract peptone (YEP) broth (Formedium LTD) and 2% of the respective carbon source (glucose (D), galactose (Gal), glycerol (G)). Alternatively, strains were grown in minimal synthetic medium containing 0.67% (w/v) yeast nitrogen base and 2% of the respective carbon sources glucose, galactose, glycerol, or lactate (SD, SGal, SG, SLac, respectively). For plates, 2% of agar was added to the medium.

### Drop dilution assay

Yeast cells were grown in YEPGal, selective SGal, or selective SG medium. During the exponential growth phase, 1 $OD_{600}$ of cells were harvested. The cells were washed with sterile water and a 10-fold dilution series (starting OD = 0.5) was prepared. Three microliters of these serial dilutions were spotted on the respective media, followed by incubation at 30 °C. Pictures were taken after different days of incubation.

### Preparation of whole cell lysates

Four $OD_{600}$ of yeast cells were harvested and washed with sterile water. Pellets were resuspended in 40 µl/$OD_{600}$ Laemmli buffer containing 50 mM DTT. Cells were lysed using a FastPrep-24 5 G homogenizer (MP Biomedicals) with three cycles of 20 s, speed 6.0 m/s, 120 s breaks, glass beads (Ø 0.5 mm) at 4 °C. Lysates were heated at 96 °C for 5 min and stored at −20 °C.

### Antibodies

Antibodies against Su9-DHFR, GFP, Sod1, Mam33, Oxa1, and Tom70 were raised in rabbits using recombinant purified proteins. For the detection of TOMM34, monoclonal antibodies were produced in mice. The anti-rabbit secondary antibody was obtained from Bio-Rad (Goat Anti-Rabbit IgG (H+L)-HRP Conjugate #172-1019). Antibodies were diluted in 5% (w/v) nonfat dry milk in 1x TBS buffer. Details about the antibodies are listed in S8 Table.

### Isolation of mitochondria

For the isolation of mitochondria, cells were grown in SGal medium to mid-log phase. Cells were harvested (2,000*g*, 5 min, 20 °C) and after a washing step, cells were treated for 10 min with 2 ml/g wet weight MP1 buffer (10 mM Tris pH unadjusted, 100 mM DTT) at 30 °C. After washing with 1.2 M sorbitol, cells were resuspended in 6.7 ml/g wet weight MP2

buffer (20 mM KPi buffer pH 7.4, 1.2 M sorbitol, 3 mg/g wet weight zymolyase 20T from Seikagaku Biobusiness) and incubated for 1 h at 30 °C. Spheroplasts were collected via centrifugation at 4 °C and resuspended in ice-cold homogenization buffer (13.4 ml/g wet weight) (10 mM Tris pH 7.4, 1 mM EDTA pH 8, 0.2% fatty acid-free BSA, 1 mM PMSF, 0.6 M sorbitol). Spheroplasts were disrupted by 10 strokes with a cooled glass potter. Cell debris was removed via centrifugation at 1,500$g$ for 5 min. To collect mitochondria, the supernatant was centrifuged at 12,000$g$ for 12 min. Mitochondria were resuspended in 1 ml of ice-cold SH buffer (0.6 M sorbitol, 20 mM Hepes pH 7.4). The Spectrophotometer/Fluorometer DS-11 FX+ (DeNovix) was used to determine the protein concentration and purity. Mitochondria were diluted to a protein concentration of 10 mg/ml.

### In vitro import into mitochondria

For the synthesis of [35]S-methionine-labeled proteins in reticulocyte lysate, the TNT Quick Coupled Transcription/Translation Kit from Promega was used. Fifty micrograms mitochondria were taken in import buffer (500 mM sorbitol, 50 mM Hepes pH 7.4, 80 mM KCl, 10 and 2 mM $KH_2PO_4$), 2 mM ATP and 2 mM NADH. As a negative control, 2% (v/v) VAO (55.6 µg/ml Valinomycin, 440 µg/ml Antimycin A, 850 µg/ml Oligomycin in ethanol (>99.8%, p.a.)) was added to the import buffer instead of ATP and NADH. If not indicated otherwise, the mixtures were incubated for 10 min at 30 °C and 650 rpm. The import reaction was started by the addition of 1% (v/v) reticulocyte lysate. Samples were taken after the indicated time points, and the reaction was stopped by a 1:10 dilution in ice-cold SH buffer (0.6 M sorbitol, 20 mM HEPES/KOH pH 7.4). Samples were supplemented with 100 µg/ml PK, and after incubation for 30 min on ice, 2 mM PMSF was added to inhibit the protease. The samples were centrifuged for 15 min at 30,000$g$ and 4 °C. The mitochondria were washed with 500 µl SH/KCl-buffer (0.6 M sorbitol, 20 mM HEPES/KOH pH 7.4, 150 mM KCl) containing 2 mM PMSF. After centrifugation for 15 min at 30,000g and 4 °C, mitochondria were resuspended in reducing sample buffer and resolved via SDS-PAGE.

If preincubation was performed prior to the import reaction, the reticulocyte lysate was preincubated at a 1:100 dilution in import buffer containing either BSA (3 or 30 mg/ml), purified TOMM34 (3 mg/ml), or the respective buffer as a control. After 10 min of preincubation, mitochondria were added to initiate the import reaction as described above.

### Aggregation assay of precursor proteins in reticulocyte lysate

[35]S-methionine-labeled proteins were synthesized in reticulocyte lysate as described above. After centrifugation (15 min, 50,000$g$, 4 °C), the supernatant was transferred into pre-cooled Eppendorf tubes. The lysate was diluted 1:100 into import buffer containing either 30 mg/ml BSA, 3 mg/ml purified TOMM34, or only the corresponding buffer without protein. After incubation at 30 °C for 0, 2, 5, or 10 min, the samples were placed on ice and subsequently centrifuged (15 min, 30,000g, 4 °C). The pellets were resolved in Laemmli buffer and analyzed via SDS-PAGE.

### Protein purification

The vector pDest15-N-His$_6$-GST-tev-TOMM34 was prepared using Gateway cloning technology. The coding sequence for TOMM34, with a TEV cleavage site at the N-terminus, was cloned into the pDest15 vector, which contains an N-terminal hexahistidinyl glutathione S transferase (His$_6$-GST) affinity tag. The vector was transformed into *E. coli* BL21 (DE3) RIPL cells for protein expression. Cells were grown in LB medium at 37 °C with shaking at 150 rpm until an OD$_{600}$ of 0.5 was reached. At this point, protein expression was induced by adding 1 mM IPTG, and the culture was incubated for an additional 3 h at 30 °C. Bacteria were harvested by centrifugation at 5,000$g$ for 15 min at 4 °C. The cell pellet was resuspended in chilled lysis buffer containing 25 mM Tris pH 7.5, 500 mM NaCl, 1 mg/ml lysozyme, and 1 mM PMSF. Cell lysis was facilitated by sonication on ice for 15 min (10 s on, 50 s off). The lysate was cleared by centrifugation at 10,000 $g$ for 30 min at 4°C. The supernatant was filtered through a 0.45 µm syringe filter (TPP) and applied to a GSTrap 5 ml FF column (Cytiva) using an ÄKTA Purifier chromatography system (Cytiva). After loading the sample, the column was washed

with equilibration buffer (25 mM Tris pH 7.5, 500 mM NaCl), and GST-TOMM34 was eluted using an elution buffer containing 20 mM reduced glutathione in the equilibration buffer. Eluted fractions were resolved by SDS-PAGE and stained with Coomassie Brilliant Blue. The protein fractions were flash-frozen in liquid nitrogen and stored at −80 °C.

### Dot plot

Peptides representing the presequences of Oxa1, Mdl2, Mam33, Atp5, Cox4, Cox51, Mas2, Atp25, Mrp20, Fum1, and Arg,5,6 with a length of 20 amino acid residues each (amino acid frame shifted by 3 amino acids from one spot to the following spot), were synthesized by a ResPep SL (Intavis, Cologne) fully automated peptide synthesizer on a derivatized cellulose membrane via their C-terminal ends according to the manufacturer instructions by solid phase peptide synthesis [79]. The cellulose membrane was incubated with methanol for 1 min at room temperature, subsequently washed twice for 1 min with $H_2O$, and equilibrated in binding buffer (150 mM Tris-HCl, pH 7.5, and 150 mM NaCl, 0.1% Triton X-100) for 2 h. After this, the membrane was blocked for 1 h in 5% milk in TBS (10 mM Tris-HCl, pH 7.4, and 150 mM NaCl) and then incubated overnight at 4 °C with 0.5 µM of recombinantly GST-TOMM34 dissolved in binding buffer. The membrane was washed three times for 10 min with binding buffer and was subjected to immunoblotting against GST.

### Fluorescence microscopy

Cells were grown to mid-log phase, and 1 $OD_{600}$ was harvested via centrifugation. Cell pellets were resuspended in 30 µl PBS. Three microliters were pipetted onto a glass slide and covered with a cover slip. Manual microscopy was performed using a Leica Dmi8 Thunder Imager with an HC PL APO100×/1,44 Oil UV objective and Immersion Oil Type A 518 F. GFP was excited by light with a wavelength of 475 nm. Further processing of images was performed with the software LAS X and Fiji/ImageJ.

### ClarioStar measurement

Cells were grown to mid-log phase in SLac + 0.2% glucose medium, and 4 $OD_{600}$ of a yeast culture were harvested by centrifugation (5,000$g$, 5 min). The pellet was resuspended in 400 µl SLac complete medium. Next, 100 µl (=1 $OD_{600}$) were transferred to a black flat-bottomed 96-well plate in technical triplicates. The plate was centrifuged at 30$g$ for 5 min to sediment the cells, and fluorescence (excitation 485 nm, emission 530 nm) was measured using a ClarioStar Fluorescence plate reader (BMG-Labtech, Offenburg, Germany). Yeast cells not expressing any fluorescent protein were used for background subtraction. Each strain was measured in biological triplicates.

### Flow cytometry

For fluorescence intensity measurements, yeast cells were either grown to mid-log phase in YEPD or SLac + 0.2% glucose medium. One $OD_{600}$ of cells were harvested, and cell pellets were resuspended in 500 µl PBS. The fluorescence intensity of 100,000 cells per strain was measured using the Attune Flow Cytometer (Thermo Fisher). Each strain was measured in biological triplicates. Data analysis was performed using the FlowJo software, version 10.

### Sample preparation and mass spectrometric identification of proteins

Co-immunoprecipitation followed by mass spectrometry was used for the identification of interactors of in vitro synthesized precursor proteins. For protein synthesis in reticulocyte lysate the TNT Quick Coupled Transcription/Translation System (Promega) was used and performed according to the manufacturer's protocol employing 1 mM unlabeled methionine. Anti-Su9-DHFR antibody (Herrmann lab) was pre-coupled to activated Protein A beads (Amintra Protein A Resin #APA0100) in IP buffer (10 mM This/HCl pH 7.5, 150 mM NaCl, 0.5 mM EDTA pH 8.0, 1 mM PMSF). Beads were washed once with IP buffer. Reticulocyte lysates were incubated with antibody-bound beads in IP buffer tumbling end-over-end for 15 min

at 4 °C. After discarding the supernatant, the beads were washed twice with wash buffer (50 mM Tris pH 7.5, 150 mM NaCl, 5% glycerol). For on-bead digestion, elution buffer I (2 M urea, 50 mM Tris pH 7.5, 1 mM DTT, and 5 ng/µl trypsin (Promega, #V5111)) were added and samples were incubated for 1 h at 20 °C. The supernatant was transferred into a fresh Eppendorf tube. For a second elution step, elution buffer II (2 M urea, 50 mM Tris pH 7.5, 5 mM chloroacetaldehyde, 5 ng/µl trypsin) was added to the beads, and the samples were incubated for 30 min at 20 °C. The supernatant of the first and second elution steps were combined and incubated overnight in the dark at 37 °C. Peptides were acidified to a pH < 2 using Tris-fluoracetic acid and desalted on in-house-prepared StageTips containing Empore $C_{18}$ disks [80]. StageTips were activated with 100 µl methanol and twice with 100 µl buffer A (0.1% formic acid). Acidified peptides were loaded onto the StageTips and washed with 100 µl buffer A. Peptides were eluted by the addition of 40 µl buffer B (80% acetonitrile, 0.1% formic acid in MS grad water) and dried using a speed vac. Samples were resolubilized in 9 µl buffer A and 1 µl buffer A* (2% acetonitrile, 0.1% tri-flouracetic acid in MS grad water).

For the analysis of whole cell proteomes, the respective yeast strains were cultured in YEPGal at 30 °C. For repression of the Cns1 expression, 10 µg/ml Doxyxycline hyclate (DOX) (Sigma, D9891-5G) was added for 30 h. Ten $OD_{600}$ of cells were harvested (5,000 g, 5 min, 20 °C) and washed with sterile water. Pellets were snap frozen in liquid nitrogen and stored at −80 °C for further analysis. For the analysis via MS, the samples were prepared according to a published protocol with minor adaptations [81]. Cell lysates were prepared in 200 µl lysis buffer (6 M guanidinium chloride, 10 mM TCEP-HCl, 40 mM chloroacetamide, 100 mM Tris pH 8.5) using a FastPrep-24 5 G homogenizer (MP Biomedicals) with 3 cycles of 20 s, speed 8.0 m/s, 120 s breaks, glass beads (Ø 0.5 mm) at 4 °C. Samples were heated for 10 min at 96 °C and afterwards centrifuged twice for 5 min at 16,000 g. In between, the supernatant was transferred to fresh Eppendorf tubes to remove all remaining glass beads. Protein concentrations were measured using the Pierce BCA Protein Assay (Thermo Scientific, #23225). For protein digestion, 25 µg of protein was diluted 1:10 with LT-digestion buffer (10% acetonitrile, 25 mM Tris pH 8.8). Trypsin (Sigma-Aldrich #T6567) and Lys-C (Wako #125-05061) were added to the samples (1:50 w/w). Samples were incubated overnight at 37 °C and 500 rpm. The pH of the samples was adjusted to pH < 2 with trifluoroacetic acid (10%) and samples were centrifuged for 3 min at 16,000 g and 20 °C. Desalting/mixed-Phase cleanup was performed using 3-layer SDB-RPS stage tips (cat 2241). Samples were dried down in speed-vac and resolubilized in 9 µl buffer A++ (buffer A (0.1% formic acid) and buffer A* (2% acetonitrile and 0.1% trifluoracetic acid) in a ratio of 9:1).

## Mass spectrometry and data analysis

Peptides were separated with an EASY-nLC 1200 ultra-high-pressure liquid chromatography system connected in-line to a Q-Exactive HF Mass Spectrometer (Thermo Fisher Scientific). Chromatography columns (50 cm, 75 mm inner diameter) were packed in house with ReproSil-Pur C18-AQ 1.9 µm resin (Dr. Maisch GmbH). Peptides were loaded in buffer A (0.1% formic acid) and eluted with a non-linear gradient of 5%–60% buffer B (0.1% formic acid, 80% acetonitrile) at a rate of 250 nl/min over 90 min for immunoprecipitations and 180 min for whole proteome samples. Column temperature was maintained at 60 °C. Data acquisition alternated between a full scan (60 K resolution, maximum injection time of 20 ms, AGC target of 3e6) and 15 data-dependent MS/MS scans (15 K resolution, maximum injection time of 80 ms, AGC target of 1e5). The isolation window was 1.4 *m/z*, and normalized collision energy was 28. Dynamic exclusion was set to 20 s.

For co-immunoprecipitation, raw mass spectrometry data was processed using MaxQuant (2.0.1.0). Peak lists were searched against a Uniprot database (UP000001811_9986) and the sequences Atp5-, Oxa1-, and Su9 presequences fused to DHFR, alongside 262 common contaminants using the Andromeda search engine. A 1% false discovery rate was applied for both peptides (minimum length: 7 amino acids) and proteins. The "Match between runs" (MBR) function was activated, with a maximum matching time window of 0.7 min and an alignment time window of 20 min. Relative protein quantities were calculated using the MaxLFQ algorithm, requiring a minimum ratio count of two. The calculation of iBAQ intensities was enabled.

The protein groups identified in each mass spectrometry data set were processed and analyzed in parallel using the R programming language (R version 4.2.2; R Core Team (2022). R: A language and environment for statistical computing. R

Foundation for Statistical Computing, Vienna, Austria. Available online at https://www.R-project.org/.). First, the MaxQuant output data was filtered to remove contaminants, reverse hits, proteins identified by site only as well as proteins that were identified in less than three replicates ($N = 3$). This resulted in 1,200 robustly identified protein groups whose label-free quantification (LFQ) intensities were log2-transformed, batchcleaned using limma [82], and further normalized using variance-stable normalization [83]. Lastly, missing values were imputed by sampling $N = 3$ values from a normal distribution (seed = 90,853,375) and using them whenever there are no valid values in a triplicate of a condition. Different for each data set, the mean of this normal distribution corresponds to the 1% percentile of LFQ intensities. Its standard deviation is determined as the median of LFQ intensity sample standard deviations calculated within and then averaged over each triplicate. Proteins were tested for differential expression using the limma package for the indicated pairwise comparison of samples, and a Benjamini–Hochberg procedure was used to account for multiple testing [84]. All relevant test results are listed in S3 Table.

Principal component analysis was carried out for each data set using the package *pcaMethods* [85] on the processed and standardized LFQ intensities of those protein groups with an ANOVA F-statistic *p*-value < 0.05 between all replicate groups to filter for proteins with a discernible degree of variance between conditions.

The protein groups identified in each whole cell mass spectrometry data set were processed and analyzed in parallel using the R programming language (R version 4.2.2; R Core Team (2022). R: A language and environment for statistical computing. R Foundation for Statistical Computing, Vienna, Austria. Available online at https://www.R-project.org/.). First, the MaxQuant output data was filtered to remove contaminants, reverse hits, proteins identified by site only as well as proteins that were identified in less than four replicates ($N = 4$). This resulted in 3,500 robustly identified protein groups whose LFQ intensities were log2-transformed and normalized using variance-stable normalization [82]. Lastly, missing values were imputed and tested for differential expression as for co-immunoprecipitation. All relevant test results are listed in S4 Table.

Principal component analysis was carried out as for co-immunoprecipitation.

For gene ontology (GO) enrichment analysis, proteins with smaller than −0.5 log2 fold change were used as target set and analyzed by using the GOrilla tool (http://cbl-gorilla.cs.technion.ac.il/) with all quantified proteins as background set (S5 Table).

## Bioinformatic analysis of mitochondrial presequences

For the bioinformatic analysis, a list of previously identified presequences [7] was adapted. Only presequences with a TargetP score > 0.3 were considered for the analysis. Duplicates were removed by only keeping the presequence, which was identified most often (column: times identified); if the abundance was the same, the longer presequence was selected. Proteins for which no presequence was identified were removed. The presequence for Oxa1 was added manually to the list. For the presentation of data in figures, we followed previously published recommendations [86].

## Physiochemical embedding of MTS

For the *in silico* comparison of different MTS, we mimicked the experimental approach by combining different MTS with the sequence of DHFR, excluding MTS of length one, which solely consist of methionine. The engineered protein sequences were embedded using the Ankh-base protein language model [87], an encoder-only model based on the T5 architecture and pretrained on the UniRef50 dataset, chosen for its high performance and relatively low hidden dimensions (768). We discarded the DHFR embeddings, extracted the MTS embeddings, and mean-pooled them to generate single feature vectors of length 768. UMAP was applied for dimensionality reduction and interpretability, transforming the data into a two-dimensional space [88]. We utilized the Python UMAP library with manual hyperparameter optimization to ensure effective separation, resulting in the following parameters: n_neighbors = 15, min_dist = 0.2, spread = 2.0, learning_rate = 0.5, and n_epochs = 1,000. Handling non-linearly separable data, clustering of similar MTS sequences was performed by spectral clustering, selected for its non-assumptive approach to cluster shape and size. The optimal number

of clusters, determined via the elbow criterion and Calinski-Harabasz score, was seven. We executed spectral clustering using the scikit-learn Python library, adjusting the gamma parameter to 1.1 and increasing iterations from 10 to 20 for enhanced consistency [89].

### Statistical analysis of different MTS sups

The individual length distributions for each cluster label were compared using the Kruskal–Wallis test to assess significant differences among clusters, with a $p$-value cutoff of 0.05. For clusters showing significant differences, Dunn's post hoc test was conducted to identify which clusters differed from the overall distribution, with p-values adjusted using the Benjamini/Hochberg method. The Kruskal-Wallis test, Dunn's post hoc test, and Benjamini/Hochberg FDR correction were performed using the SciPy, scikit-learn, and statsmodels libraries, respectively [90]. We applied the same procedure to evaluate whether specific properties of the proteins from which the MTS originated were enriched in any clusters. The properties analyzed included Gravy Score, Abundance Median, Aromaticity Score, Codon Bias, Protein Length, Molecular Weight, Aliphatic Index, Instability Index, and Protein Half-life, all sourced from the YeastMine online service [91]. Additionally, we used DeepStabP to estimate the melting temperature of each protein as a proxy for protein stability [92].

### Evolutionary comparison of MTS across species

To compare MTS across species, we collected MTS data from various organisms using UniProt [93]. The following filters were applied: transit peptide mitochondrion, any evidence type, protein existence with evidence at the protein level, and an annotation score of 5. We further refined the dataset by removing all MTS with a length of one or unspecified lengths. Subsequently, we combined this curated dataset with the original yeast MTS dataset. The combined dataset was then subjected to the same physiochemical embedding process previously outlined, utilizing the Ankh-base protein language model for embedding, UMAP for dimensionality reduction, and spectral clustering for grouping similar sequences.

### Validation of presequence classification with published proteomics data

To assess the functional significance of presequence groups across multiple mitochondrial perturbation conditions, we used previously published proteomics data sets for which the effects of different mutations and stress conditions on the abundance of mitochondrial proteins in yeast had been analyzed [20,22,55,94–96], all represented as $\log_2$-transformed fold changes relative to a control state. To this end, we used published proteomics datasets. For details, see S2 Table. For each condition, the absolute mean of these $\log_2$ fold changes was calculated per presequence cluster to identify functionally important clusters. We then applied a permutation test (99,999 iterations, *$p$*< 0.05) against randomly assembled protein groupings to determine which clusters exhibited more extreme mean absolute log fold changes compared to non-essential clusters. For cross-dataset comparability (S1C Fig), the absolute mean values were further standardized via $z$-score normalization. The S2 Table provides complete results, including cluster-wise absolute means with corresponding p-values and analyses focused on exclusively positive or negative log fold changes within each cluster. All computational analyses were performed using Python 3.10.0 with the polars 0.20.16 data processing library.

## Supporting information

**S1 Fig. Classification of presequences. (A)** Schematic visualization of a multi-step workflow for LLM-based physiochemical embedding, created to compare different MTS in silico. The workflow begins by replacing the N-terminal methionine of the cytosolic protein dihydrofolate reductase (DHFR) with the targeting sequences. The engineered proteins are then processed through a protein language model to embed the sequences in high-dimensional space. Subsequently, the DHFR sequence is cleaved off, leaving only the embedded MTS sequence. To mitigate any bias related to sequence length, a mean-pooling layer is applied. Finally, Uniform Manifold Approximation and Projection (UMAP) is used to project

the embedded sequences into two-dimensional space for interpretation. This workflow is repeated for each MTS, with UMAP applied to all embedded sequences simultaneously to facilitate grouping of similar vectors. **(B)** Evolutionary Comparison of MTS Across Species. Comparison of clusters identified in the S. cerevisiae datasets (220 MTS—colored) with an expanded dataset containing 1,450 MTS from proteins of a comprehensive set of organisms (S1 Table) reveals the evolutionary robustness of the different groups found in yeast. **(C)** Shown are the consistencies of changes upon different perturbations for representatives of the different groups of presequences. The absolute mean values of the changes within a group were calculated. The significance indicates how likely it is to obtain the respective mean value by chance, considering the cluster size, the respective condition, and the respective cluster. Thus, high values mean that proteins with specific presequence types are more consistently affected than it would be expected if the distribution was random. The data underlying the graphs shown in the figure can be found in S1 Data.
(PDF)

**S2 Fig. Presequences can be sorted into seven distinct groups on basis of their sequence.** List of the gene names that correspond to the embedded MTS sorted according to the identified cluster.
(PDF)

**S3 Fig. All tested presequences depend on the levels of the membrane potential and ATP. (A–C)** Mitochondria were isolated from wild type, Δcox18, and Δatp6 mutants. The Δatp6 strain was grown in the absence of arginine to ensure the presence of the mitochondrial genome. This mutant contained the *ARG8* gene at the position of the mitochondrially encoded ATP6 locus [76]. For panel C, the strain MR6 was used as a corresponding wild type of the Δatp6 mutant. The indicated proteins were radiolabeled and imported as described for Fig 2A. Import efficiencies were quantified and are shown in relation to the import efficiency into wild-type mitochondria. The data underlying the graphs shown in the figure can be found in S1 Data.
(PDF)

**S4 Fig. Fluorescence intensities of the IQ-Comp assay can be measured by fluorescence spectroscopy and microscopy. (A)** Scheme of the in vivo import assay IQ-Compete. **(B, F)** Microscopy images of the respective strains. See legend to Fig 3D for details. **(C, D)** Fluorescence intensities of the depicted strains were measured using a multiplate fluorescence spectrometer (Clariostar, BMG Labtech). Shown are mean values and standard deviations of three biological replicates. **(E)** The western blot of the samples shown in Fig 4B was probed with antibodies against DHFR for detection of the uTEV-DHFR fusion proteins. The data underlying the graphs shown in the figure can be found in S1 Data.
(PDF)

**S5 Fig. Presequences mediate the association with the cytosolic chaperone TOMM34. (A, B)** Su9-DHFR was imported into isolated mitochondria with or without urea-denaturation as described for Fig 5A and 5B. **(C)** Protein enrichment in Su9-DHFR samples relative to the DHFR control. See Fig 6B for details. **(D)** Specific enrichment (log2-fold change) of specific chaperones with Atp5-DHFR and Oxa1-DHFR relative to the DHFR control. **(E)** Coomassie-stained gels showing N-GST-TOMM34 expression in *Escherichia coli* under uninduced (−IPTG) and induced (+IPTG) conditions. Protein solubility was assessed by separating soluble (S) and insoluble (P) fractions. Recombinant GST-TOMM34 was purified using GSTrap chromatography, with flow-through (FT) and elution fractions shown. **(F)** Peptides of 20 residues representing the presequences of the indicated proteins were spotted onto a cellulose membrane. The individual peptide sequences were moved by a three-residue window (1-20, 4-23, 7-26, etc.). The membrane was incubated with purified recombinant GST-TOMM34. The membrane was washed, blocked with milk powder, and probed with an anti-GST antibody. **(G)** Model depicting the position of specific TOMM34 binding sites in presequences. The data underlying the graphs shown in the figure can be found in S1 Data.
(PDF)

**S6 Fig. In yeast, the cytosolic co-chaperone Cns1 facilitates mitochondrial biogenesis. (A)** Principal component analysis of the proteome data of the indicated strains. PC1 impressively shows the strong effect of Cns1 depletion which is not compensated by TOMM34 expression (PC2). **(B)** List of most severely depleted mitochondrial proteins in Cns1-depleted cells. **(C, D)** The expression of TOMM34 in the Cns1-depleted cells has only a very minor effect on mitochondrial proteins. The data underlying the graphs shown in panes A, C, and D can be found in S4 Table. **(E)** The expression of TOMM34 does not complement the growth defect of Cns1-depleted cells, whereas the expression of Cns1 from a plasmid complements the mutant.
(PDF)

**S1 Raw Images. Raw gels.**
(PDF)

**S1 Data. Supplementary data.**
(XLSX)

**S1 Table. List of organisms used for the bioinformatic analysis of presequences.** Refers to S1B Fig.
(PDF)

**S2 Table. Combined proteomics data sets used for the bioinformatic analysis of the respective changes of presequence-containing proteins.** Refers to S1C Fig.
(XLSX)

**S3 Table. Results of the mass spectrometric identification of presequence-bound proteins.** Refers to Fig 6A–6C.
(XLSX)

**S4 Table. Results of the mass spectrometric identification of the whole cell proteomes of wild-type and Cns1-depleted cells.** Refers to Fig 8E and 8F.
(XLSX)

**S5 Table. GO term enrichment. Shows the GO terms of protein groups that were affected upon depletion of Cns1.** Refers to Fig 8F.
(XLSX)

**S6 Table. Yeast strains used in this study.**
(XLSX)

**S7 Table. Plasmids used in this study.**
(XLSX)

**S8 Table. Antibodies used in this study.**
(XLSX)

## Acknowledgments

We thank Sabine Knaus for technical assistance, Johannes Buchner for the Cns1 depletion strain, and Jean-Paul di Rago for the Δatp6 mutant.

## Author contributions

**Conceptualization:** Saskia Rödl, Johannes M. Herrmann.

Data curation: Saskia Rödl, Yasmin Hoffman, Felix Jung, Oliver Šimončík, Martin Jung, Markus Räschle, Zuzana Storchová.

Formal analysis: Saskia Rödl, Yasmin Hoffman, Annika Nutz, Martin Jung, Markus Räschle, Petr Muller, Zuzana Storchová.

Funding acquisition: Timo Mühlhaus, Johannes M. Herrmann.

Investigation: Saskia Rödl, Yasmin Hoffman, Johannes M. Herrmann.

Methodology: Saskia Rödl, Yasmin Hoffman, Annika Egeler, Timo Mühlhaus, Johannes M. Herrmann.

Resources: Timo Mühlhaus, Johannes M. Herrmann.

Software: Felix Jung.

Validation: Saskia Rödl, Johannes M. Herrmann.

Visualization: Saskia Rödl, Yasmin Hoffman, Felix Jung.

Writing – original draft: Saskia Rödl, Johannes M. Herrmann.

Writing – review & editing: Yasmin Hoffman, Felix Jung, Annika Egeler, Annika Nutz, Oliver Šimončík, Martin Jung, Markus Räschle, Petr Muller, Zuzana Storchová, Timo Mühlhaus.

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
