## [Editor Report · Decision Letter 0]

Dear Dr Herrmann,

Thank you for submitting your manuscript entitled "A priority code in presequences: mitochondrial targeting signals assign specific import characteristics to precursor proteins" for consideration as a Research Article by PLOS Biology. Please note that I am currently handling your manuscript whilst my colleague Ines is away from the office on holiday for the next few weeks.

Your manuscript has now been evaluated by the PLOS Biology editorial staff, as well as by an academic editor with relevant expertise, and I am writing to let you know that we would like to send your submission out for external peer review.

Once your full submission is complete, your paper will undergo a series of checks in preparation for peer review. After your manuscript has passed the checks it will be sent out for review. To provide the metadata for your submission, please Login to Editorial Manager (https://www.editorialmanager.com/pbiology) within two working days, i.e. by Aug 07 2024 11:59PM.

Kind regards,

Richard

Richard Hodge, PhD

rhodge@plos.org

On behalf of:

Ines Alvarez-Garcia, PhD

---

## [Decision Letter · Decision Letter 1]

Dear Dr Herrmann,

Thank you for your patience while your manuscript entitled "A priority code in presequences: mitochondrial targeting signals assign specific import characteristics to precursor proteins" was peer-reviewed at PLOS Biology. Please also accept my apologies for the delay in providing you with our decision. Your manuscript has been evaluated by the PLOS Biology editors, an Academic Editor with relevant expertise, and by three independent reviewers.

The reviews are attached below. As you will see, the reviewers find the work potentially interesting, but they also raise a substantial number of important concerns that would need to be addressed before we consider the manuscript for publication. Each of the reviewers make many reasonable suggestions for additional experiments to address the concerns they raise, which for the most part are quite similar. You should focus on those shared criticisms and perform the experiments that better address the open questions. We do think that the most important criticism is the weak connection between the machine learning approach and the actual experimental results. In addition, the choice of the substrates should be better explained, and you should address the role of TOMM34 and its putative functional analogue in yeast and provide a more detailed analysis, especially considering that the in vivo analyses have been done in a hybrid system using mammalian reticulocyte lysate and isolated yeast mitochondria. The ER-SURF pathway should be also discussed.

Given the extent of revision that would be needed, we cannot make a decision about publication until we have seen the revised manuscript and your response to the reviewers' comments. Your revised manuscript would need to be seen by the reviewers again, but please note that we would not engage them unless their main concerns have been addressed.

We appreciate that these requests represent a great deal of extra work, and we are willing to relax our standard revision time to allow you 6 months to revise your study. Please email us (plosbiology@plos.org) if you have any questions or concerns, or envision needing a (short) extension.

**IMPORTANT - SUBMITTING YOUR REVISION**

3. Resubmission Checklist

a) *PLOS Data Policy*

b) *Published Peer Review*

Sincerely,

Ines

--

Ines Alvarez-Garcia, PhD

Senior Editor

PLOS Biology

Reviewers' comments

Rev. 1:

In this study, Rödl and colleagues conduct a thorough investigation into the features of mitochondrial presequences, aiming to unravel the unique mechanisms behind protein import into mitochondria. The research introduces new techniques and offers several strong contributions. However, some aspects of the data presentation and writing suggest that the paper needs refinement.

The authors propose a priority code model in which a subset of mitochondrial precursors utilizes more potent presequences to recruit cytosolic targeting factors. The paper begins with a broad perspective and an extensive dataset, which is commendable. However, as the study progresses, the sample size of presequences used to develop and support the conclusions diminishes. For instance, the cytosolic factor TOM34, which is suggested to play a key role, is shown to interact with only one presequence (OXA1), but the conclusion is broad suggesting all strong presequences are using TOM34. This limitation undermines the findings of the work.

Some more specific comments can be found below.

[1] The study presents an elegant machine learning approach to categorize mitochondrial presequences, successfully identifying seven distinct groups (A-G). However, in Figure 2, the authors analyze the features of nine proteins shown in Figure 1B, but they do not select representatives from each of the proposed categories (with categories D, F, and G notably absent). This omission diminishes the perceived significance of the machine learning approach, as it appears the authors are simply using the precursors they had before conducting the categorization. Would make sense to include precursor proteins across their identified categories.

[2] The precursors suggested to have stronger import capabilities (Oxa1, Su9, Mdl2, and Pim1) also possess longer presequences. On page 4, it is stated that "presequences of this group carry structural determinants that endow them with an increased import competence." Could this simply be attributed to the length of the presequences? The authors should explore this possibility by categorizing the precursors identified in their machine learning approach according to presequence length and investigating import efficiency as a correlation of presequence length.

[3] The conclusions drawn from Figure 2 and Supplementary Figure 2 need to be toned down. The authors state on page 5 "The import efficiency of all proteins into these compromised mitochondria was consistently lower (Fig. 2E, F, S2B, C)." However, while the atp5 mutant shows a broad defect, the cox18 mutant is less pronounced. There is minimal quantification for these experiments and the authors should therefore moderate the language used in these conclusions and provide quantifications for some precursors across the different categories to support their claims.

[4] Pre-incubation experiment (Fig 5C and D) where it is concluded: longer preincubations do not affect the import of Atp5-DHFR, but strongly that of Oxa1-DHFR (Fig. 5C, D), suggesting that the cytosolic factor might be released upon prolonged incubation in the buffer. Can the authors show that this is not due to the OXA1 precursor aggregating and coming out of solution and there is therefore less protein available for import.

[5] The relevance of Figure 5 and just looking at ATP5 and OXA1 really diminishes the significance of the paper. If the conclusions of this section are to be taken seriously it need to be assessed across a larger spectrum of substrates.

[6] Tom34 enrichment with OXA1. It would be good to assess the relevance of this finding to the overall picture of this study. Is this an OXA1 specific phenotype, or something specific for precursors with longer presequences. This experiment should be undertaken with some stronger and weaker presequences to support the authors claims.

[7] The results and discussion sections include speculative conclusions about Cns1 functioning similarly to Tom34 in yeast, despite the fact that Tom34 could not rescue the growth phenotype associated with Cns1 deficiency in yeast. This speculation regarding Cns1 should be tempered in the discussion. The authors should consider synthesizing the OXA1 presequence, along with appropriate controls, in yeast translation extracts to directly assess whether Cns1 interacts with these presequences.

Rev. 2:

Many mitochondrial proteins contain an N-terminal cleavable presequence that carries a targeting signal for the mitochondrial matrix. Although these presequences generally form a positively charged amphiphilic helix, which functions as the mitochondrial targeting signal, they vary in length, amino-acid sequences, and targeting/import efficiency. In this study, Rödl et al. conducted in-silico analyses of mitochondrial presequences and successfully classified them into several categories (Groups A to G), each exhibiting different import efficiencies. Subsequently, they performed both in vitro and newly developed in vivo import assays to investigate the reasons behind these variations in import efficiency. They found that presequences of Group A proteins, such as Oxa1, exhibit higher import efficiency, prompting further investigation into the underlying mechanisms. Specifically, they sought to uncover the mechanism responsible for the higher import efficiency observed in Group A precursors. This study offers insights into the longstanding question of presequence properties, potentially uncovering previously elusive mechanisms that regulate import efficiency. However, the authors' interpretation of the reasons for the different import efficiencies among the various groups of presequences lacks sufficient experimental evidence. Consequently, the manuscript's current content is not compelling enough to convince me and appears too premature for publication.

1. The authors found that the differences in import efficiency among the different presequence groups were not affected by reduced energization, i.e. by the lowered membrane potential across the inner membrane or decreased ATP levels in the mutant cells such as delta-cox18 or delta-atp6 cells (Fig. 2E and F). Based on these observations, the authors concluded that "the import efficiency differences are not due to different intramitochondrial requirements but presumably by a step upstream, i.e. by the targeting to and interaction with mitochondrial surface receptors" (page 5). However, this conclusion has shortcomings. It is already known, in contrast to the interpretation here, that different presequeces can be affected by different levels of the membrane potentials when modulated by different concentrations of CCCP (Martin et al., JBC 266, 1991). Indeed, even the Oxa1-DHFR import here looks less sensitive to the decrease in the membrane potential than Atp5-DHFR (Fig. 2E). Therefore, the authors should perform in vitro import assays with CCCP titration to validate their findings.

2. A potential problem the authors should address is the complexity of the protein import process, which involves multiple steps and various factors and mechanisms. The authors state that "these factors might support the import in two ways: first, by increasing the targeting efficiency to the mitochondrial surface, and second, by the unfolding of precursor proteins to facilitate their translocation through the mitochondrial protein translocases" (page 8). However, this statement is speculative and lacks experimental support, necessitating further experimental assessment. If the former is correct, the targeting step should be the rate-limiting one in the overall import process, which could be experimentally confirmed through import assays using reduced amounts of isolated mitochondria (Lithgow and Schatz (1995) J.Biol.Chem.270,14267-14269). Alternatively, if the latter is the case, the folding state of precursor proteins in complex with chaperones could be tested through protease digestion analysis.

3. They showed that the significant difference in the import efficiency between Oxa1-DHFR and other precursors was eliminated by urea denaturation of the precursor proteins prior to the import. However, since urea could affect the import machinery, the authors should try other conditions such as performing the in vitro import at a higher temperature (e.g. 37C), where the folded DHFR domain becomes more unstable, and a destabilized DHFR mutant as a passenger protein.

4. The authors also observed that prolonged preincubation of in vitro synthesized precursor proteins after dilution into the import buffer resulted in reduced import efficiency. They speculated that a cytosolic import-stimulating factor may be released during prolonged incubation in the buffer (Fig. 5C and D). However, this experiment requires a control involving prolonged incubation without dilution or dilution into a buffer containing BSA or glycerol to mimic the cytosol crowding conditions.

5. The authors synthesized Oxa1-DHFR and other precursors in reticulocyte lysate, followed by immunoprecipitation using anti-DHFR antibodies, and analyzed the co-precipitated proteins by mass spectrometry. They identified several cytosolic chaperones, such as Hsp70 and its co-chaperones, but found that TOMM34 co-immunoprecipitated specifically with Oxa1-DHFR and not with Atp5-DHFR (Fig. 6C). Pretreatment with anti-TOMM34 antibodies reduced the import efficiency of Oxa1-DHFR but had no effect on Atp5-DHFR (Fig. 6F). Based on these results, the authors speculated that Oxa1-DHFR binds to the cytosolic chaperone TOMM34, which could be a factor contributing its enhanced import efficiency (Fig. 6J). However, the roles of cytosolic chaperone or targeting factors are complex, as these proteins may interact with substrates transiently and in a redundant, multivalent manner. Several critical questions remain unanswered: what is the apparent MW of Oxa1-DHFR interacting with chaperones in reticulocyte lysate? How might nucleotide conditions (e.g. ATP depletion) and temperature affect the spectrum of proteins interacting with the model precursor proteins used in this study? What is the time course of import competency and possible aggregate formation for Oxa1-DHFR in reticulocyte lysate? The most critical point is that pretreatment of Oxa1-DHFR with anti-TOMM34 antibodies alone is insufficient to demonstrate that TOMM34 is the primary factor responsible for the higher import efficiency of Oxa1-DHFR. The authors should first deplete TOMM34 from the reticulocyte lysate using anti-TOMM34 antibodies, and then synthesize Oxa1-DHFR to test its import efficiency under these conditions.

6. They found that the import of Oxa1-DHFR was less efficient in the absence of Tom70, which functions as a docking point for the Hsp70 and Hsp90 family proteins for import. However, as Tom70 itself also has chaperone functions (Yamamoto et al., JBC 284, 31635 (2009)), Tom70 may directly interact with the Oxa1 presequence to prevent its aggregation. This possibility should be tested experimentally.

Other points

The newly proposed GFP-based in vivo import assay is interesting, but it lacks kinetic information, making it unclear which step of the import process is rate-limiting. These in vivo import assays require further characterization to be efficiently compared with in vitro import results.

Line12 in page 6 - group 1 should read group A

Rev. 3:

Mitochondrial protein biogenesis depends on the import of precursor proteins from the cytosol. Most of the precursors carry N-terminal extensions (presequences), which are cleavable targeting signals that mediate targeting to the organelle and subsequent translocation across the mitochondrial membranes. While presequences have been identified using N-proteomic approaches, a global analysis of their characteristics has not been performed so far. In the present manuscript, Rödl et al set out to fill this knowledge gap using machine learning and different import assays.

The analysis of presequence characteristics as the major mitochondrial targeting signals is of interest to the field of protein localization, and the current manuscript provides an interesting technical approach to study mitochondria protein targeting in vivo. However, the results obtained are preliminary and do not provide new biological insight. This is mainly due to the fact that the manuscript is not consistent as it seems that different ideas and data have been forced together. I have three main criticisms:

The machine learning approach yields 7 classes. While one would expect the authors to proceed with the analysis of 1-2 representatives of these new classes, they have chosen to continue with the classical presequences used before (by this they cover only four of the "new" classes), ignoring their own findings. It is therefore not clear why the machine learning approach was carried out at all.

The main finding of the manuscript is the requirement of TOMM34 for a specific precursor protein. TOMM34 was identified using a proteomics approach. Unfortunately, it is not clear why the authors focused on TOMM34, as many other proteins show a stronger enrichment (the proteomics data are only available on PRIDE, which is not user-friendly, and usually additional supplemental excel files are provided for easy assessment and evaluation of the data). While the choice of TOMM34 is not sufficiently clear, the real problem lies in the whole experimental set-up: The authors use isolated yeast mitochondria, but combine them with precursor proteins synthesized in rabbit reticulocyte lysate. It is true that this approach has been used for decades in the mitochondrial protein biogenesis field. While this may be acceptable for the study of intramitochondrial translocation processes, it is highly problematic when investigating the early cytosolic steps of mitochondrial protein targeting. Yeast extract for in vitro translation is well established and has been successfully used by other groups to study cytosolic events in mitochondrial targeting (in particular by the Rapaport lab). Here, using a rabbit translation system, the authors identify TOMM34 and assign it a role in mitochondrial targeting, despite the fact that yeast does not have a TOMM34 homologue. Therefore, there are no data on the in vivo relevance of this protein.

Finally, the authors identify the presequence of Oxa1 as the most efficient. This is curious as the group has previously published a novel protein import pathway, in which precursors are imported into mitochondria via the ER (Hansen et al., ER SURF pathway). If the import of Oxa1 is the fastest, how and why would the precursor then be re-routed via the ER? Unfortunately, the authors do not mention the ER-SURF at all in the manuscript, which is surprising given that they identified this new import pathway.

---

## [Decision Letter · Decision Letter 2]

Dear Hannes,

Thank you for your patience while we considered your revised manuscript entitled "A priority code in presequences: mitochondrial targeting signals assign specific import characteristics to precursor proteins" for publication as a Research Article at PLOS Biology. Your revised study has been evaluated by the PLOS Biology editors, the Academic Editor and two of the original reviewers. The Academic Editor has also checked your responses to Reviewer 3's comments.

The reviews are attached below. As you will see, while Reviewer 1 is now fully satisfied, Reviewer 2 has raised two remaining concerns that were not addressed. The first one refers to the fact that the in vitro import assay with reduced amounts of isolated mitochondria was not performed, as it seems to be the only way to identify factors that affect the targeting steps and not the later step of the import. The second point is about the statement saying that TOMM34 depletion from reticulocyte lysate is beyond the scope of the study, as the reviewer thinks it is not difficult to perform and it would be very useful to include the results in the manuscript.

After discussing the comments with the Academic Editor, we do agree with Reviewer 2 that the experiments mentioned seem feasible. Thus, we would like you to perform them, especially the first one, or explain more specifically the reasons for not including them.

Note that we cannot make a decision about publication until we have seen the revised manuscript and your response to the reviewer's comments. We expect to receive your revised manuscript within 3 months. Please email us (plosbiology@plos.org) if you have any questions or concerns, or would like to request an extension.

**IMPORTANT - SUBMITTING YOUR REVISION**

3. Re-submission Checklist

Sincerely,

Ines

--

Ines Alvarez-Garcia, PhD

Senior Editor

PLOS Biology

Reviewers' comments

Rev. 1:

The authors have addressed all my comments and the inclusion of new data has strengthened the manuscript overall. I recommend the manuscript proceed to publication.

Rev. 2:

The authors have addressed most of my comments, but their responses remain insufficient regarding the following points.

(1) The authors did not properly address my comment 2 ('A potential problem the authors should address is…). While they described the merit of the in vivo import analyses, my concern is why they did not perform an in vitro import assay with reduced amounts of isolated mitochondria, which is not so difficult but is the only way to reveal factors that affect the targeting step, not the later step of the import.

(2) In response to comment 5, the authors stated that TOMM34 depletion from reticulocyte lysate is beyond the scope of this study. However, this experiment is not difficult and would be worthwhile to perform.

---

## [Editor Report · Decision Letter 3]

Dear Hannes,

Thank you for your patience while we considered your new revised manuscript entitled "A priority code in presequences: mitochondrial targeting signals assign specific import characteristics to precursor proteins" for publication as a Research Article at PLOS Biology. This revised version of your manuscript has been evaluated by the PLOS Biology editors and by the Academic Editor.

Based on our Academic Editor's assessment of your revision, we are likely to accept this manuscript for publication, provided you satisfactorily address the data and other policy-related requests stated below my signature.

In addition, we would like you to consider a suggestion to improve the title:

"A protein-specific priority code in presequences determines the efficiency of mitochondrial protein import”

We expect to receive your revised manuscript within two weeks.

*Published Peer Review History*

*Press*

Sincerely,

Ines

--

Ines Alvarez-Garcia, PhD

Senior Editor

PLOS Biology

DATA POLICY:

Fig. 1C-G; Fig. 2B, E, F; Fig. 3C, E, F; Fig. 4C, D, E; Fig. 5B, D, F, H, J, L; Fig. 6A, B, C, F; Fig. 7B, E, F, H; Fig. 8D-G; Fig. S1B, C; Fig. S3B, C; Fig. S4C, D; Fig. S5B, C, D and Fig. S6A, C, D

**Please also make publicly available at this stage the data you have deposited in the ProteomeXchange Consortium via the PRIDE partner repository (PXD053210 and PXD059975).

CODE POLICY

Many thanks for submitting the raw gels for all blot and gel results. I have checked them all and I am missing the raw gel corresponding to Fig. S5E.

---

## [Editor Report · Decision Letter 4]

Dear Dr Herrmann,

Thank you for the submission of your revised Research Article entitled "A protein-specific priority code in presequences determines the efficient of mitochondrial protein import" for publication in PLOS Biology. On behalf of my colleagues and the Academic Editor, Andre Schneider, I am delighted to let you know that we can in principle accept your manuscript for publication, provided you address any remaining formatting and reporting issues. These will be detailed in an email you should receive within 2-3 business days from our colleagues in the journal operations team; no action is required from you until then. Please note that we will not be able to formally accept your manuscript and schedule it for publication until you have completed any requested changes.

PRESS

Sincerely, 

Ines

--

Ines Alvarez-Garcia, PhD

Senior Editor

PLOS Biology
